# Thought-Retriever: Don't Just Retrieve Raw Data, Retrieve Thoughts for Memory-Augmented Agentic Systems

**Tao Feng**[*]                                                          *taofeng2@illinois.edu*
*University of Illinois Urbana-Champaign*

**Pengrui Han**[*]                                                          *phan3@mit.edu*
*University of Illinois Urbana-Champaign*

**Guanyu Lin**[*]                                                          *guanyul@andrew.cmu.edu*
*Carnegie Mellon University*

**Ge Liu**                                                          *geliu@illinois.edu*
*University of Illinois Urbana-Champaign*

**Jiaxuan You**                                                          *jiaxuan@illinois.edu*
*University of Illinois Urbana-Champaign*

**Reviewed on OpenReview:** *https://openreview.net/forum?id=emCcuhtENL*

## Abstract

Large language models (LLMs) have transformed AI research thanks to their powerful *internal* capabilities and knowledge. However, existing LLMs still fail to effectively incorporate the massive *external* knowledge when interacting with the world. Although retrieval-augmented LLMs are proposed to mitigate the issue, they are still fundamentally constrained by the context length of LLMs, as they can only retrieve top-K raw data chunks from the external knowledge base which often consists of millions of data chunks. Here we propose *Thought-Retriever*, a novel model-agnostic algorithm that helps LLMs generate output conditioned on arbitrarily long external data, without being constrained by the context length or number of retrieved data chunks. Our key insight is to let an LLM fully leverage its intermediate responses generated when solving past user queries (*thoughts*), filtering meaningless and redundant thoughts, organizing them in thought memory, and retrieving the relevant thoughts when addressing new queries. This effectively equips LLM-based agents with a self-evolving long-term memory that grows more capable through continuous interaction. Besides algorithmic innovation, we further meticulously prepare a novel benchmark, AcademicEval, which requires an LLM to faithfully leverage ultra-long context to answer queries based on real-world academic papers. Extensive experiments on AcademicEval and two other public datasets validate that Thought-Retriever remarkably outperforms state-of-the-art baselines, achieving an average increase of at least 7.6% in F1 score and 16% in win rate across various tasks. More importantly, we further demonstrate two exciting findings: (1) Thought-Retriever can indeed help LLM self-evolve after solving more user queries; (2) Thought-Retriever learns to leverage deeper thoughts to answer more abstract user queries.

 ulab-uiuc/Thought-Retriever

## 1 Introduction and Related Work

Large language models (LLMs) have revolutionized AI research thanks to their powerful *internal* capabilities (Zhao et al., 2023; Wang et al., 2023) and knowledge (Peng et al., 2023a). When building LLMs, researchers

---

[*]Equal contribution. Corresponding author.

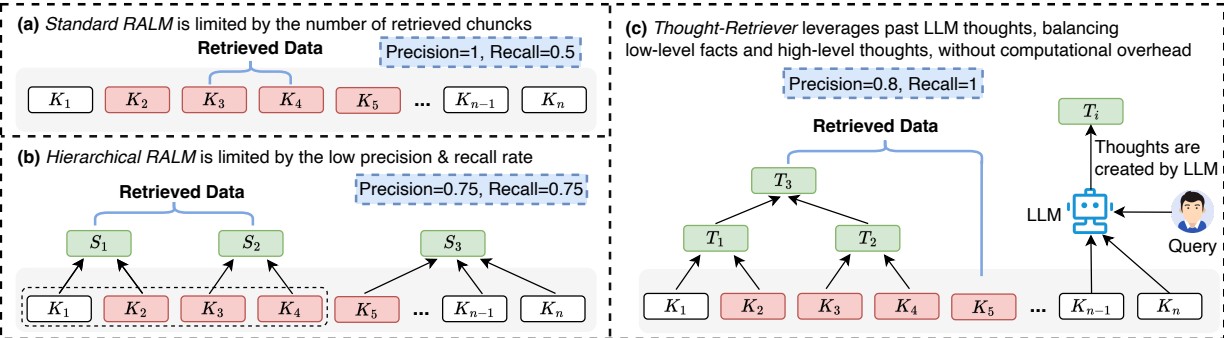

Figure 1: **Why Thought-Retriever helps**. **(a)** A standard RALM is limited by the number of retrieved chunks. The retrieved data fails to cover all the necessary data chunks (red chunks) for a user query. **(b)** A hierarchical RALM retrieves summaries $S_i$, generated independently from user queries, which could improve recall at the cost of lower precision. **(c)** Thought-Retriever leverages past LLM thoughts collected from answering user queries, with little computational overhead. Thought-Retriever balances low-level facts and high-level thoughts, leading to high precision and recall.

further expect LLMs to interact with the world by effectively incorporating the *external knowledge* as their long-term memories, *e.g.*, collected from *facts* (Sun et al., 2023) or interactions with *other AIs* (Wu et al., 2023; Kannan et al., 2023). Importantly, the scale of the external knowledge for LLMs could be arbitrarily large; ultimately, all the digitized information within our universe could serve as the external knowledge for these LLMs. In practice, when building personalized LLM applications (Bill & Eriksson, 2023) or LLM-powered domain experts (Thirunavukarasu et al., 2023; Liu et al., 2023), *e.g.*, AI doctor, the relevant external knowledge for the LLMs could also easily get extremely large, *e.g.*, billions of tokens. Therefore, our paper aims to raise attention to the pressing research question: *how to effectively and efficiently help LLMs and LLM-based agents utilize (arbitrarily) rich external knowledge as long-term memory.*

To help LLMs better incorporate external knowledge, existing research mainly falls into two categories: *long-context LLMs* and *retrieval-augmented LLMs (RALMs)*. (1) *Long-context LLMs*, such as MPT (MosaicML, 2023) and LongChat (LM-SYS, 2023), aim to expand the LLM's context window, *e.g.*, via novel training algorithms (Tay et al., 2022), inference algorithms (Xiao et al., 2023), new architectures (Peng et al., 2023b; Gu & Dao, 2023), or system optimization (Xu et al., 2023c). Although these methods improve the working memory size of LLMs, they cannot fundamentally address the issue of interacting with ultra-rich external knowledge using LLMs, since the computational complexity is often quadratic to the context length. (2) *RALMs* retrieve pertinent information from external knowledge bases using retrievers, such as BM-25 (Robertson et al., 2009), Contriever (Izacard et al., 2022), and DRAGON (Lin et al., 2023). However, these algorithms are still constrained by LLMs' context length, since they can only retrieve top-K raw data chunks from the external knowledge that fits within an LLM's context limit. (3) *Hierarchical RALMs*, *e.g.*, creating a tree-structured memory for an LLM (Chen et al., 2023). Despite its potential to help LLMs incorporate more abstract knowledge, manually summarizing closed chunks and rigidly forming a tree structure proves to be a costly and inefficient method. This approach demands significant resources and lacks the flexibility to adapt to specific inputs in LLMs. Overall, existing methods in attempting to include external knowledge for LLMs still exhibit *fundamental limitations in efficiency and effectiveness*. Meanwhile, recent LLM-based agentic systems (Park et al., 2023; Wang et al., 2024; Packer et al., 2023) also require persistent memory to maintain coherence across extended interactions. However, these frameworks typically store raw observations or unprocessed interaction logs, which are noisy and difficult to retrieve effectively. This highlights the need for a more compressed and reasoning-aware memory mechanism for agents.

Here, we propose *Thought-Retriever*, an LLM-agnostic self-evolving retrieval framework that leverages historical LLM responses to answer new queries. Our key insight is that LLM responses can be transformed into *thoughts* with little computational overhead and that the thoughts can be organized as a thought memory for the LLM to facilitate future tasks. Psychological studies (Kurzweil, 2013; Snell, 2012) support our insight, revealing that human memory is organized hierarchically, which not only aids in retrieving relevant information for problem-solving but also gradually deepens our understanding of the world through continuous processing and summarizing these interactions into complex cognitive thoughts. Notably, through

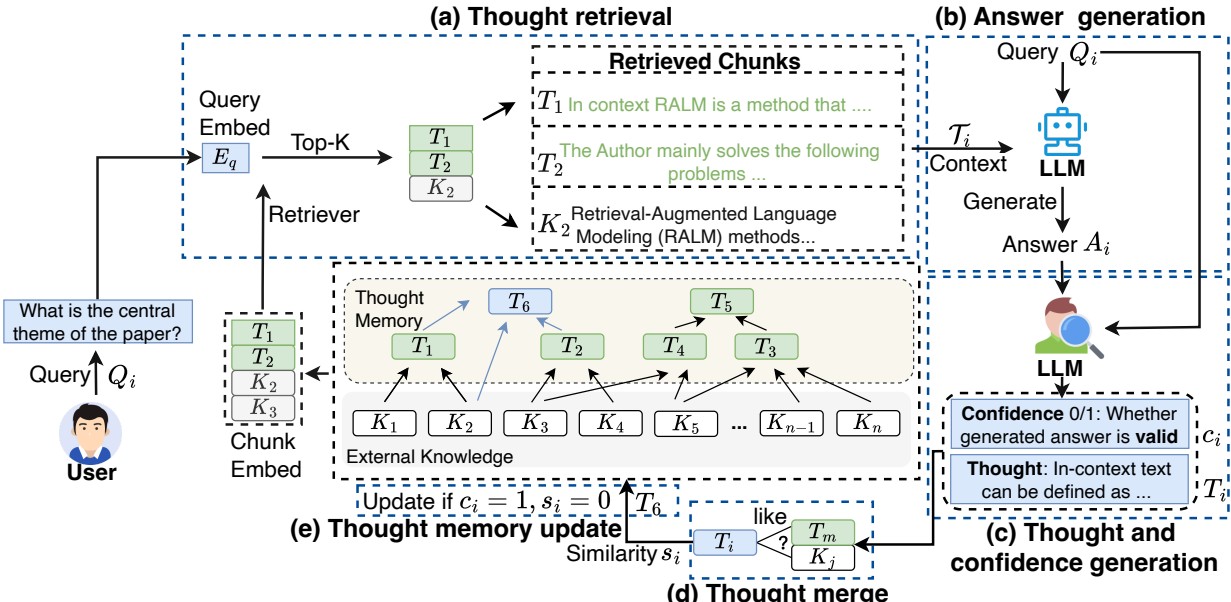

Figure 2: **Thought-Retriever Framework**. **(a) Thought retrieval:** Upon receiving a user query, Thought-Retriever retrieves top-K data chunks from the mixture of external knowledge and thought memory based on embedding similarity; (b) **Answer and confidence generation:** The LLM generates the answer for the user query based on the retrieved data chunks; (c) **Thought generation:** The LLM further generates thoughts and its confidence based on the user query and the generated answer; (d) **Thought merge:** The calculation of similarity is used to measure whether the generated thought will cause redundancy in data chunks; (e) **Thought memory update:** Meaningless and redundant thoughts are removed and the remaining *novel* thoughts are used to update the thought memory.

continuous interaction with diverse user queries, Thought-Retriever progressively generates more novel and expansive thoughts. This is achieved by organizing new data chunks from external knowledge into thoughts after addressing each query, filtering out meaningless and redundant thoughts, and ultimately incorporating high-quality thoughts into the thought memory. Therefore, Thought-Retriever gives an LLM agent the potential to utilize arbitrarily rich external knowledge as long-term memories and achieve self-evolution in capabilities.

In addition to algorithmic advancements, we also meticulously developed a novel benchmark, *AcademicEval*, which challenges an LLM to accurately utilize extensive context to answer queries based on real-world academic papers. Our comprehensive experiments on AcademicEval and two additional datasets confirm that Thought-Retriever significantly surpasses state-of-the-art baselines, achieving an average increase of at least 7.6% in F1 score and 16% in win rate across various tasks. Furthermore, we present two intriguing discoveries: (1) Thought-Retriever can indeed facilitate the self-evolution of an LLM after addressing more user queries; (2) Thought-Retriever is capable of harnessing deeper insights to respond to more abstract user queries.

In summary, our *main contributions* are as follows: **(1)** Thought-Retriever framework enables an LLM agent to efficiently and effectively utilize external knowledge as long-term memory, and further allowing it to self-evolve through continuous interactions. **(2)** AcademicEval, a real-world benchmark for testing LLM's understanding of ultra-long context. **(3)** Thought-Retriever consistently outperforms all state-of-the-art retrieval-augmented and long-context baselines and presents exciting new findings.

## 2 Thought-Retriever: Effectively Equip LLMs with External Knowledge

### 2.1 Preliminaries

An agent's *external knowledge* base $\mathcal{K} = (K_1, K_2, ..., K_n)$ consists of $n$ data chunks. Instead of treating model outputs merely as transient responses, we formally define a *thought* ($T_i$) as a persistent, query-driven cognitive unit derived from the interaction between a query $Q$ and retrieved context $\mathcal{K}_i$. Formally, a Thought $T_i$ is a validated abstraction $T_i = \text{Abstract}(Q, A, \mathcal{K}_i)$ that satisfies three key properties: (1) **Query-Conditioned**: it captures the logical link between a specific query and the data, unlike static summaries; (2) **Abstractive**: it distills a coherent knowledge point from the raw conversation; and (3) **Validated**: it passes a confidence check ($c_i$) and a novelty check ($s_i$) to ensure it is meaningful and non-redundant.

We define the *immediate source* of a thought $T_i$ as the set of retrieved items $\mathcal{R}_i$ (which may contain both raw chunks and other thoughts) used to generate it. To rigorously trace information provenance, we define the *root source* mapping $\hat{O}(\cdot)$ recursively:

1. **Base Case:** For a raw data chunk $K \in \mathcal{K}$, the root source is the chunk itself: $\hat{O}(K) = \{K\}$.

2. **Recursive Step:** For a thought $T$ generated from a retrieved set $\mathcal{R}$, the root source is the union of the root sources of its components: $\hat{O}(T) = \bigcup_{r \in \mathcal{R}} \hat{O}(r)$.

**Example:** Consider a thought $T_{new}$ generated based on an existing thought $T_{old}$ and a raw chunk $K_3$ (i.e., $\mathcal{R} = \{T_{old}, K_3\}$). If $T_{old}$ was originally derived from raw chunks $\{K_1, K_2\}$, then the root source of the new thought is $\hat{O}(T_{new}) = \hat{O}(T_{old}) \cup \hat{O}(K_3) = \{K_1, K_2, K_3\}$. This ensures that we can always verify the factual grounding of any high-level thought.

### 2.2 Motivating Examples

To measure how effectively an LLM agent can utilize external knowledge, we propose to extend the retrieval metric, precision, and recall, with the root source mapping $\hat{O}(\cdot)$. Assuming that answering a user query $Q_{\text{think}}$ requires a set of data chunks $\mathcal{K}_i \in \mathcal{K}$, and an LLM's response is $T_i$. We have $\text{Precision} = \frac{|\mathcal{K}_i \cap \hat{O}(T_i)|}{|\hat{O}(T_i)|}, \text{Recall} = \frac{|\mathcal{K}_i \cap \hat{O}(T_i)|}{|\mathcal{K}_i|}$. As a motivating example, in Figure 1, we assume $\mathcal{K}_i = \{K_2, K_3, K_4, K_5\}$ is required to answer a user query and an LLM can only fit 2 data chunks in its context window. A standard RALM (Figure 1(a)) can achieve perfect precision by retrieving the correct data chunks; however, it has a lower recall since it does not have the context window to hold all the relevant data chunks.

To address the limited context window of RALM, researchers (Chen et al., 2023) proposed hierarchical RALMs (Figure 1(b)), where similar data chunks are summarized into $S_i$ via LLM as a preprocessing step. However, the tree-structured summary structure is rigid, since the summaries $S_i$ are *static compressions* generated independently from user queries. In contrast, our Thought-Retriever generates *dynamic thoughts* that are conditioned on specific user interactions, allowing the memory to evolve based on actual data usage patterns. In Figure 1(b), ideally, chunks $\{K_2, K_3\}$ and $\{K_4, K_5\}$ should be grouped together to answer the user query, where Precision = 1, Recall = 1 could be achieved; however, the tree construction happened before user query, and the generated tree fail to adapt to the diverse future user query.

To stress the above limitations of existing RALMs, as is shown in Figure 1(c), we propose the Thought-Retriever that leverages past LLM thoughts and balances low-level facts and high-level thoughts to answer user queries. In real-world applications, user queries are often sufficiently diverse, leading to numerous diverse thoughts to meet the demands of new user queries. This valuable observation differentiates Thought-Retriever from existing tree-structured RALMs: (1) Thought-Retriever offers a more flexible structure of thoughts that depends on past user queries, and (2) the thoughts leveraged by Thought-Retriever are byproducts from the standard RALM response, making it easy to implement and bringing little computational overhead.

---

**Algorithm 1** Thought-Retriever Inference Algorithm

---

**Input:** User queries $\mathcal{Q}$, external knowledge $\mathcal{K}$, thought memory $\mathcal{T}$, language model $L$, retriever $R$ and threshold of similarity $\epsilon$.

**Output:** Answers to user queries $\mathcal{A}$, updated thought memory $\mathcal{T}$.

1: $\mathcal{A} \leftarrow \{\}$
2: **for** $Q_i \in \mathcal{Q}$ **do**
3:     $\mathcal{T}_i \leftarrow R(Q_i, \mathcal{K} \cup \mathcal{T})$                                      ▷ Thought retrieval
4:     $A_i \leftarrow L(Q_i, \mathcal{T}_i)$                                           ▷ Answer generation
5:     $\mathcal{A} \leftarrow \mathcal{A} \cup A_i$
6:     $T_i, c_i \leftarrow L(Q_i, A_i)$                           ▷ Thought and confidence generation
7:     $s_i \leftarrow \mathbf{1}_{\{\exists j, m \; ; sim(T_i, K_j / T_m) \geq \epsilon\}}$                             ▷ Thought merge
8:     $\mathcal{T} \leftarrow \mathcal{T} \cup T_i$, if $c_i = 1, s_i = 0$                     ▷ Thought memory update
9: **end for**
10: **return** $\mathcal{A}, \mathcal{T}$

---

## 2.3 Thought-Retriever Framework

**Method Overview.**    Figure 2 offers an overview of the proposed Thought-Retriever framework, which serves as a general-purpose memory module for LLM-based agents and consists of four major components: (1) **Thought retrieval**, where data chunks from external knowledge and thought memory are retrieved; (2) **Answer generation**, where an LLM generates the answer for the user query based on the retrieved data chunks; (3) **Thought and confidence generation**, where an LLM further generates thought and its confidence in validation to avoid hallucination based on the user query and the generated answer; (4) **Thought merge**, where similarity is calculated to measure whether generated thought will cause redundancy in data chunks; (5) **Thought memory update**, where meaningless and redundant thoughts are removed; the thought memory is updated with the remaining *novel* thoughts, rather than adopting all the *new* thoughts. We summarize the pipeline of Thought-Retriever in Algorithm 1, whose details are shown as follows. Detailed prompts for this section can be found in Appendix A.2.

**Thought Retrieval.**    After receiving a user query $Q_i$, Thought-Retriever $R$ retrieves relevant information $\mathcal{T}_i$ from external knowledge $\mathcal{K}$ and previously generated thought memory $\mathcal{T}$ via embedding similarity ranking. This process is formulated as $\mathcal{T}_i \leftarrow R(Q_i, \mathcal{K} \cup \mathcal{T})$.

**Answer Generation.**    Based on the retrieved information $\mathcal{T}_i$, we design a prompt to combine $\mathcal{T}_i$ and user query $Q_i$ and feed the prompt to an LLM $L$ to get the answer $A_i$. It can be articulated as $A_i \leftarrow L(Q_i, \mathcal{T}_i)$.

**Thought and Confidence Generation.**    We can generate thoughts via LLM $L$ using the obtained answer $A_i$ and its query $Q_i$ (an example is shown in Table 5). However, meaningless thoughts during the generation process may cause hallucinations for LLM and harm performance since some queries may be irrelevant to the external knowledge and thought memory. To solve this issue, we design a special prompt so that LLM $L$ can generate thought $T_i$ and thought quality confidence $c_i$ based on the user's query $Q_i$ and corresponding answer $A_i$. This can be described as $T_i, c_i \leftarrow L(Q_i, A_i)$. Crucially, this step differentiates a *thought* from a standard *response* ($A_i$). While $A_i$ aims to satisfy the user's immediate request, $T_i$ is explicitly generated to distill the reasoning logic into a "coherent knowledge point" (as shown in the prompt in Figure 3) for the system's long-term memory. Specifically, $c_i$ is a discrete binary value, where 1 indicates that the generated thought $T_i$ is meaningful, and 0 indicates that it is meaningless or hallucinated. This confidence generation is also validated through our experiment in Sec E.1.

**Thought Merge.**    Redundant thoughts may cause LLM to retrieve duplicate information, which is also harmful to the performance of LLM. Therefore, we calculate the similarity between the generated thought $T_i$ and data chunks $(T_m, K_j)$ to measure whether the generated thought $T_i$ will cause redundancy in data chunks. Instead of using complex notation, we implement this as a direct threshold check: we compute the

> Input: Given question:{question}, given answer:{context}. Based on the provided question and its corresponding answer, perform the following steps:
>
> Step 1: Determine if the answer is an actual answer or if it merely indicates that the question cannot be answered due to insufficient information. If the latter is true, just output '0' without any extra words, otherwise output '1'.
>
> Step 2: If it is a valid answer, succinctly summarize both the question and answer into a coherent knowledge point, forming a fluent passage.

Figure 3: **Thought and Confidence Generation Prompt.** This prompt is used for Thought and Confidence Generation as described in Section 2.3. It evaluates whether the answer is valid and meaningful, and then summarizes the query and answer into a thought.

maximum embedding similarity between $T_i$ and all existing items in the memory. If this maximum score exceeds the threshold $\epsilon$, we flag the thought as redundant (setting $s_i = 1$); otherwise, we consider it novel (setting $s_i = 0$). Here, the embedding similarity is calculated based on Contriever (Izacard et al., 2022) (same as retriever used elsewhere in Thought-Retriever).

**Thought Memory Update.** The confidence of thought quality $c_i$ and the similarity $s_i$ determine whether the newly generated thought should be updated into the thought memory $\mathcal{T}$. Here, we design that if the LLM is confident about its answer and the generated thought is not redundant, where $c_i = 1, s_i = 0$, $\mathcal{T}$ will be updated.

## 3 AcademicEval: New Benchmark for Long-Context LLM Understanding

Current benchmarks for assessing LLM long-context memory utilization involve tasks such as question-answering, long-context summarization, and classification. Despite being well-constructed, they are limited in flexibility and real-world impact and are costly to acquire due to human labeling. To address these issues, we introduce an innovative benchmark, **AcademicEval**, based on academic papers from arXiv updated daily. *AcademicEval* comes with two datasets: *AcademicEval-abstract* and *AcademicEval-related*. We also launched a public platform that will enable users to easily create similar datasets or utilize LLMs for academic tasks (see details in Appendix H). In addition, the detailed dataset introduction and usage instructions can be found in Appendix A.1 and A.2 respectively.

**(1) AcademicEval-abstract.** This dataset focuses on the summarization of single (*Abstract-single* in Table 1) or multiple (*Abstract-multi* in Table 1) academic papers. The LLM is presented with one or more papers with the abstract and conclusion sections removed and is tasked with writing an abstract. For *Abstract-single*, the generated abstract is directly compared with the paper's original abstract. For *Abstract-multi*, the generated abstract is compared with a summary of abstracts from all the provided papers, which is generated by an expert LLM as a label. **(2) AcademicEval-related.** This dataset (*Related-multi* in Table 1) introduces a challenging task for assessing an LLM's ability to understand the connections between heterogeneous segments of its long-context memory. The task is to write a related work section based on the title and abstract of a target paper. The LLM needs to use the title and abstract as the query to retrieve memory chunks to complete this task. To be specific, memory chunks depict the abstracts of several papers (each memory chunk corresponds to the abstract of a paper), where some papers are cited in the related work section of the target paper, while others are randomly sampled from the same broader field. The generated related work is then compared to the original related work of the target paper for evaluation.

Table 1: **Overview of Datasets:** task types, average length, and number of cases.

| Dataset | Task Type | Avg. len | Cases |
|---|---|---|---|
| **AcademicEval** | | | |
| Abstract-single | Single Sum | 8,295 | 100 |
| Abstract-multi | Multi Sum | 33,637 | 30 |
| Related-multi | Multi Related | 22,107 | 30 |
| **Public Datasets** | | | |
| Gov Report | Single QA | 8,910 | 100 |
| WCEP | Multi QA | 8,176 | 30 |

## 4 Experiment

### 4.1 Experiment Setup

**Additional Datasets.** Besides AcademicEval, we further evaluate Thought-Retriever against state-of-the-art baselines on two public datasets. (1) **GovReport** (Cao & Wang, 2022): This dataset comprises 19,466 reports and associated labels prepared by government research agencies to verify if the LLM is capable of extracting salient words and useful information from a single lengthy governmental document. (2) **WCEP** (Ghalandari et al., 2020): This dataset contains 10,200 entries, each containing multiple news articles associated with an event sourced from the Wikipedia Current Events Portal. It requires the LLM to understand and extract useful information from a cluster of documents. Table 1 summarizes the statistics for all the datasets. Additionally, we discuss the computational details in Appendix N. It is important to note that the current experimental validation is conducted in English. However, the Thought-Retriever framework is inherently language-agnostic. We acknowledge that exploring its application to multilingual settings and low-resource languages is a promising direction, which we plan to investigate in future work to further demonstrate the framework's broad applicability.

**Baselines.** To gain a comprehensive understanding of our thought retriever's performance on LLM long-term memory tasks, we have adopted several baselines. All experiments with these baselines are conducted under the same LLM: Mistral-8x7B with LLM context length of 4,096 (Jiang et al., 2024). Note that we set chunk size=500, K=8, $\epsilon = 0.85$, and maximum context length=2,000 tokens for all RALMs. We employ Contriever (Izacard et al., 2022) as our primary retriever due to its unsupervised design, which provides strong zero-shot performance across diverse domains without requiring labeled training data (unlike supervised alternatives like DPR). This aligns with our goal of building a general-purpose framework. Its empirical superiority over other retrievers is further verified in our ablation study in Sec 4.5. First, we consider 2 heuristic-based retrievers **BM25** (Robertson et al., 2009) and **TF-IDF** (Ramos et al., 2003). Second, we select 4 deep learning-based retrievers: **Contriever** (Izacard et al., 2022), **DPR** (Karpukhin et al., 2020), **DRAGON** (Lin et al., 2023), and the state-of-the-art decoder-only embedding model **Qwen3-Embed-8b** (Zhang et al., 2025), which leverages the Qwen3 foundation model for enhanced multilingual text understanding and retrieval. Third, to evaluate advanced retrieval strategies, we include **IRCoT** (Trivedi et al., 2023a) , an iterative method that interleaves retrieval with chain-of-thought reasoning to dynamically guide the information-seeking process. Fourth, we employ **RECOMP** (Xu et al., 2023a), a context compression technique that generates textual summaries of retrieved documents to reduce computational cost while maintaining information density. Fifth, we consider full context window baselines with document truncation **Full Context (left)** (Chen et al., 2023) and **Full Context (right)** (Chen et al., 2023). Lastly, we selected two long-context LLMs **OpenOrca-8k** (Mukherjee et al., 2023) and **Nous Hermes-32k** (Shen et al., 2023). Note that we do not compare with MEMWALKER (Chen et al., 2023), since it is costly to run and cannot scale to tasks with many data chunks. The details of baselines can be found in Appendix B.

**Evaluation Metrics.** Our evaluation approach encompasses both traditional metric and AI-based assessments: (1) **F1** (Lin, 2004): This metric computes the semantic similarity between the generated text and the ground truth reference through ROUGE-L (F1). An F1 score closer to 1 indicates a higher alignment

Table 2: **Thought-Retriever consistently outperforms all the baselines in fact retrieval datasets**. **Bold** and underline denote the best and second-best results. F1 score evaluates the similarity with the ground truth, higher is better. Win Rate represents the frequency with which a method's response is preferred over Thought-Retriever by the evaluator. The 50% entry for Thought-Retriever serves as the reference point (a tie); consequently, for other baselines, a win rate lower than 50% indicates that they are outperformed by Thought-Retriever. Note that the maximum context length is 2,000 tokens for all retriever-based methods and Thought-Retriever employs Contriever as its retriever.

| Type | AcademicEval | | | | | | Public | | | |
|---|---|---|---|---|---|---|---|---|---|---|
| Dataset | Abstract-single | | Abstract-multi | | Related-multi | | Gov Report | | WCEP | |
| Method | F1 | Win Rate | F1 | Win Rate | F1 | Win Rate | F1 | Win Rate | F1 | Win Rate |
| BM25 | 0.212 | 7% | 0.232 | 7% | 0.203 | 40% | 0.211 | 30% | 0.178 | 31% |
| TF-IDF | 0.202 | 4% | 0.225 | 4% | 0.207 | 40% | 0.195 | 35% | 0.223 | 34% |
| Contriever | 0.242 | 13% | 0.232 | 15% | 0.201 | 35% | 0.223 | 40% | 0.211 | 40% |
| DPR | 0.206 | 4% | 0.226 | 4% | 0.196 | 30% | 0.188 | 20% | 0.201 | 33% |
| DRAGON | 0.236 | 7% | 0.226 | 8% | 0.208 | 30% | 0.210 | 40% | 0.231 | 35% |
| Qwen3-Embed-8b | 0.245 | 28% | 0.240 | 20% | 0.211 | 35% | 0.229 | 42% | 0.235 | 44% |
| IRCoT | 0.243 | 25% | 0.235 | 18% | 0.209 | 33% | 0.225 | 41% | 0.233 | 42% |
| RECOMP | 0.237 | 7% | 0.202 | 8% | 0.198 | 10% | 0.215 | 35% | 0.205 | 33% |
| Full Context (left) | 0.118 | 2% | 0.155 | 0% | 0.193 | 13% | 0.234 | 45% | 0.207 | 35% |
| Full Context (right) | 0.118 | 1% | 0.149 | 0% | 0.188 | 8% | 0.220 | 40% | 0.210 | 41% |
| OpenOrca-8k | 0.175 | 20% | 0.135 | 3% | 0.135 | 13% | **0.244** | 41% | 0.169 | 30% |
| Nous Hermes-32k | 0.247 | 30% | 0.204 | 7% | 0.183 | 15% | 0.238 | 37% | 0.214 | 37% |
| **Thought-Retriever** | **0.290** | **50%** | **0.275** | **50%** | **0.216** | **50%** | 0.232 | **50%** | **0.238** | **50%** |

with the reference text, signifying the better quality of the generated content. (2) **Win Rate**: Alongside F1, we incorporate feedback from the AI evaluator for a more comprehensive assessment. Here, we choose Qwen1.5-72B-chat as our AI evaluator, since it has superb alignment with human preference[1]. This evaluation process involves presenting various responses to the LLM evaluator, who then ranks the quality of the responses. The percentage represents the frequency of a response being chosen over our thought retriever. A rate below 50% suggests that our thought retriever is outperforming the compared baseline.

## 4.2 Retrieve Context from Factual Knowledge

This section is to verify the performance of Thought-Retriever when the external knowledge comes from interaction with facts. We report the performance of our model and baselines in Table 2. Major observations are as follows:

First, in both *AcademicEval* and public benchmarks, Thought-Retriever significantly outperforms most baselines on two metrics. For example, it achieves an average increase of at least 7.6% in F1 score and 16% in win rate across all datasets. This suggests that thoughts formed through interaction with the environment can effectively enhance an LLM's performance in different tasks. Moreover, the comparison and analysis of abstracts generated by different methods on the Abstract-single task (Appendix F) also verify the effectiveness of Thought-Retriever. Second, we observe that the performance of methods that use the entire text directly have many features on two different benchmarks differs greatly, which contain Full Context baselines and long-context LLMs baselines. However, the performance of retriever-based methods is stable across two benchmarks. This is due to two reasons: (1) AcademicEval is a more challenging benchmark. It contains "multi-modal" information, such as tables, different chapters, different symbol formats, etc. Directly putting this complicated information in a context makes it difficult for the LLM to process and analyze. For retriever-based methods, they extract the most important information for respond the query from the entire memory, so they can filter out the influence of some redundant information and get better results; (2) Some long-context LLMs may have continuously train on the public benchmarks, which causes the leak of the label and the overfitting of the model. In contrast to this, AcademicEval is a good benchmark for evaluating the zero-shot performance of LLM and has no risk of label leakage and overfitting. Since the benchmark is formed using papers from arXiv, it is dynamic and always up-to-date, benefiting from the continuous

---

[1]https://qwenlm.github.io/blog/qwen1.5/

Table 3: **Thought-Retriever can help the LLM quickly learn from other LLMs**. Retriever-origin is the golden setting that retrieves original facts, others are comparative settings without original facts.

| Setting | Retriever-origin | Response-direct | Retriever-other | Thought-Retriever |
|---------|------------------|-----------------|-----------------|-------------------|
| F1 | 0.25 | 0.19 | 0.22 | 0.24 |

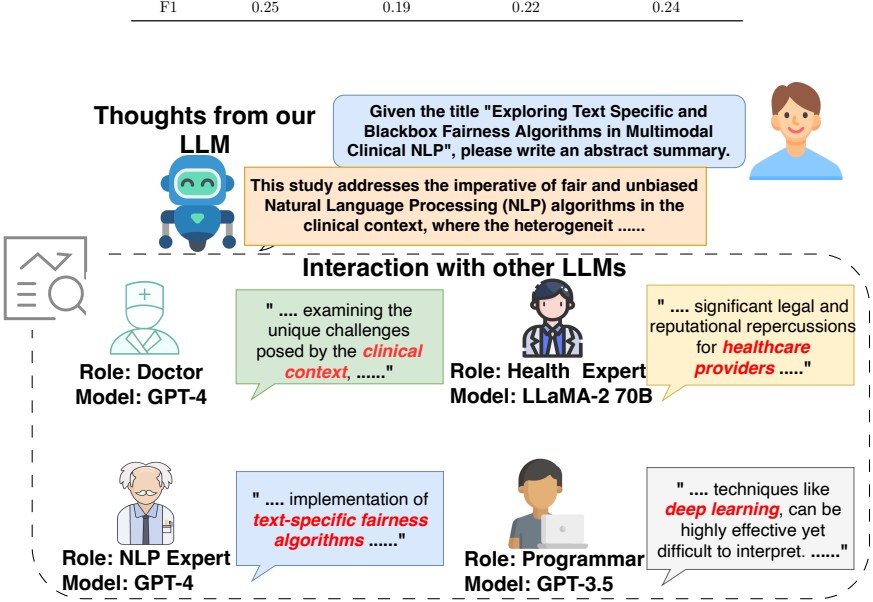

Figure 4: **Thoughts from other LLMs help respond without fact.** It presents an illustrative example in which our LLM communicates with four other LLMs, each an expert in a different field. These expert LLMs are assigned specific roles (e.g., doctor) with different background knowledge. Our LLM is then able to rapidly learn from their thoughts and incorporate them as external knowledge.

publication of new papers. We further show the comparison between Thought-Retriever and other SOTA baselines in Appendix L.

## 4.3   Retrieve Context Generated from other LLMs

Forming thoughts can be a lengthy process. When a new LLM lacks relevant memory or external knowledge, it is challenging to develop high-quality thought memories from scratch. Consequently, we aim to investigate whether Thought-Retriever can help the LLM quickly learn from other LLMs that have already formed expert knowledge. To answer this question, we design an experiment on Abstract-single, and the goal of the LLM is to write an abstract summary based on its title. Our LLM builds its memories based on interaction with other LLMs, which include different roles of an LLM or different LLMs, as shown in Fig 4. To verify the effectiveness of Thought-Retriever under this setting, we design four different comparison settings: (a) **Retriever-origin** retrieves knowledge based on the original context of the papers and then uses this knowledge to respond to queries, which serves as a golden setting; (b) **Response-direct** feeds the query directly to the LLM to get the responses; (c) **Retriever-other** let other LLMs provide some relevant data based on a query, then uses this knowledge as raw memories of our LLM, and finally retrieves and gets response based on retriever; (d) **Thought-Retriever** utilizes Thought-Retriever to construct thought memories then retrieve thoughts for responding queries based on the setting of **Retriever-other**. We evaluate with metric F1 in 30 cases of Abstract-single, and the results shown in Table 3 demonstrate that the rank of them from good to bad is: Retriever-origin, Thought-Retriever, Retriever-other, Response-direct. Moreover, the response quality of Thought-Retriever is very close to that of Retriever-origin. These observations verify the effect and efficiency of Thought-Retriever when learning from other LLMs. Further results on QA and Reasoning tasks Li et al. (2023) can be found in Appendix I.

### 4.4 New Findings from Thought-Retriever

**(1) Thought Retriever learns to leverage deeper thoughts to answer more abstract user queries.** We conduct a case study to explore the relationship between the abstraction levels of queries and the retrieved information. Specifically, we created a set of questions with varying levels of abstraction and ranked them according to their abstraction level using an expert LLM (exact queries can be found in Appendix E).

To quantify the depth of retrieved information, we introduce a formal measure of *Abstraction Level*, denoted as $\mathcal{L}(x)$. The calculation follows a recursive definition:

- **Base Case (Raw Data):** For any raw data chunk $K \in \mathcal{K}$ from the external knowledge base, we assign a baseline abstraction level of $\mathcal{L}(K) = 1$.

- **Recursive Step (Thoughts):** For a generated thought $T$, let $\mathcal{R}_T$ be the set of items (either raw chunks or existing thoughts) retrieved to generate it. The abstraction level of $T$ is calculated as the average level of its sources plus one:

$$\mathcal{L}(T) = 1 + \frac{1}{|\mathcal{R}_T|} \sum_{r \in \mathcal{R}_T} \mathcal{L}(r) \tag{1}$$

This recursive formula ensures that a thought derived solely from raw data has a level of 2 (i.e., $1+1$), while a thought synthesized from other high-level thoughts will achieve a strictly higher abstraction score (e.g., $> 2$), reflecting deeper cognitive processing.

As shown in Figure 5, where the y-axis represents the abstraction level of the question and the x-axis represents the average abstraction level of all information retrieved by our method. It can be observed that more abstract questions tend to retrieve information with higher abstraction levels.

**(2) Thought-Retriever helps LLM self-evolve after solving more user queries - a new type of scaling law.** To investigate the relationship between the performance of Thought-Retriever and the number of thoughts, we design an experiment using varying numbers of thoughts on Abstract-multi and Related-multi of AcademicEval. As depicted in Figure 6, there is a distinct trend of increasing F1 scores correlating with the growing number of thoughts, which indicates improved performance. Therefore, more interactions with the users enable Thought-Retriever to assist LLM agents in self-evolving and developing deeper understandings, demonstrating a new type of scaling law (Kaplan et al., 2020) for agentic memory. We also verify the diversity of thoughts and effectiveness of thought filtering mechanisms in Appendix K and M.

### 4.5 Ablation Study

We conduct a series of experiments to investigate the impact of various retrievers. (1) **w/wo TF-IDF/DPR/DRAGON**: In these variants, we replace the retriever (Contriever) in our method with other representative retrievers to assess their effectiveness compared to our current retriever. (2) **w/wo NousHermes**: Here, we utilize NousHermes to construct the thoughts. (3) **w/o Filter**: We remove the confidence generation and thought merge in our framework to assess the importance of filtering meaningless and redundant thoughts. We report the evaluation results on Abstract-single and Abstract-multi datasets in Figure 7. **These comparisons clearly show that our method consistently outperforms all the variants**, suggesting that Contriever is most suitable for Thought-Retriever, and filtering meaningless and redundant thoughts can bring great improvement to the performance.

### 4.6 Qualitative Analysis based on Precision and Recall

In our motivation example in Sec 2.2, we highlighted where traditional methods struggle with recall and precision. Here, using the Related-multi dataset, we show that Thought-Retriever outperforms other baselines in balancing both metrics. In the experiment, the abstracts of the real citations are regarded as ground truth. We aimed to assess how well different retrievers could retrieve information to cover the ground truth,

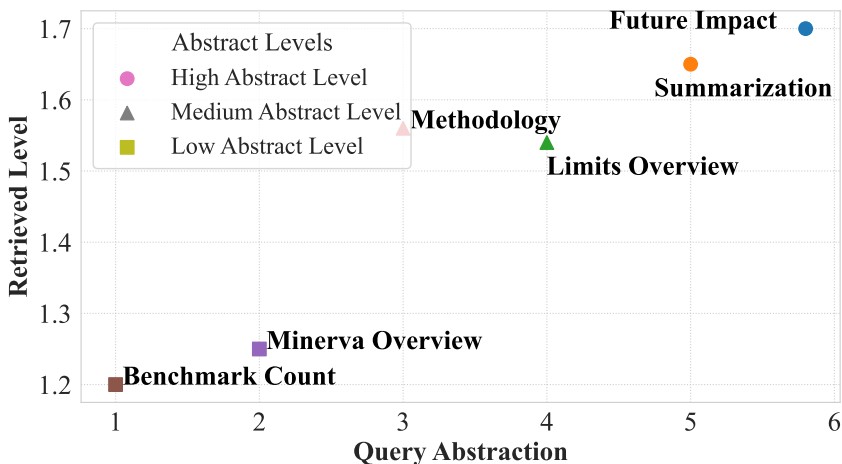

Figure 5: **Deeper thoughts help abstract queries.** This figure illustrates the correlation between six questions, categorized by their level of abstraction as evaluated by expert LLM (x-axis), and the abstraction level of the corresponding retrieved information (y-axis). The questions are grouped into three categories: high abstraction (top 2 questions), medium abstraction, and low abstraction. Keywords from each question are displayed next to their corresponding data points for clarity.

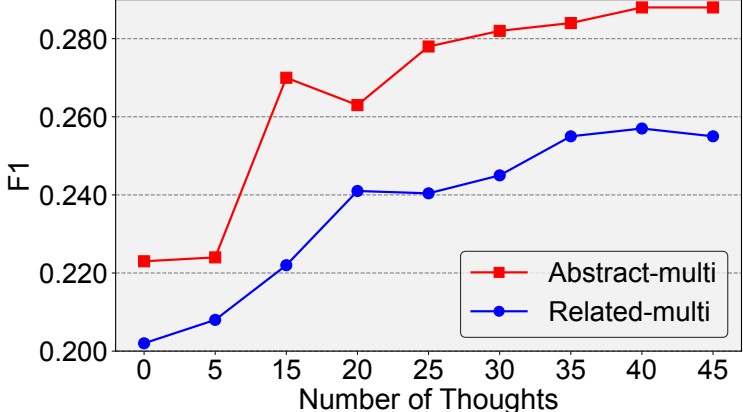

Figure 6: **Thought-Retriever can indeed help LLM self-evolve after solving more user queries. It illustrates that the performance of LLM across two datasets shows an upward trend as the number of thoughts increases.**

given the limitation of retrieving only 8 chunks of information at a time. We plotted the findings in Figure 8, where the x-axis is the recall value and the y-axis represents the precision. It can be observed that all traditional retrieval methods displayed significantly low recall values. This is primarily attributed to the top-K retrieval limit since K=8 is far less than the number of ground truth citations. In comparison, Thought-Retriever demonstrates a notable improvement in recall value. This is because it leverages thoughts that are constructed from multiple papers, thereby allowing Thought-Retriever to achieve a much higher recall. More importantly, the Thought-Retriever also exhibits moderately high precision compared to other retrievers. This suggests that, despite a minor trade-off, Thought-Retriever does not significantly compromise its ability to retrieve the most relevant information.

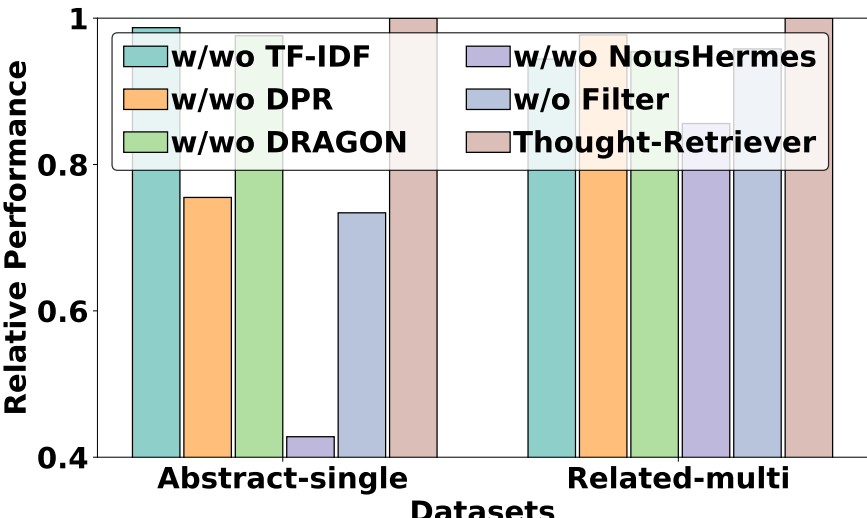

Figure 7: **Contriever and thoughts filtering are suitable for Thought-Retriever.** Ablation study of 6 methods on two datasets helps us decide on important components of Thought-Retriever.

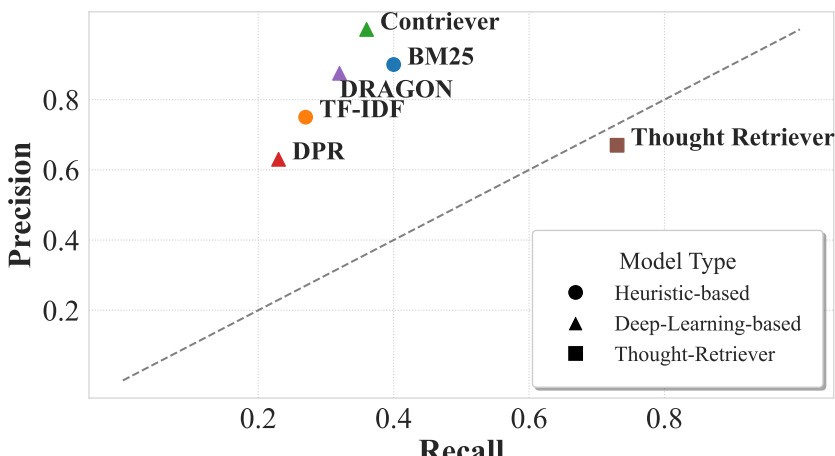

Figure 8: **Thought-Retriever performs better in balancing recall and precision.** The dotted line indicates the exact balance between precision and recall. The closer the dotted line is, the better the balance is.

## 5 Additional Related Works

**LLM Agent Memory and Experiences.** Autonomous agents rely on persistent memory mechanisms to maintain coherence over extended interactions, effectively emulating the concept of experience replay in continual learning. Prominent frameworks like Generative Agents (Park et al., 2023) and Voyager (Wang et al., 2024) utilize memory streams or skill libraries to store past observations and behaviors, enabling agents to evolve based on historical experiences. Similarly, MemGPT (Packer et al., 2023) manages context as an operating system manages hierarchical memory. While long-context LLMs attempt to address this by encompassing all history within a massive context window, they suffer from quadratic computational complexity and the lost-in-the-middle phenomenon, often failing to effectively utilize distant information. Our approach bridges these paradigms but with a distinct advantage: instead of storing raw observations (as in standard agents) or processing exhaustive raw contexts (as in long-context models), Thought-Retriever stores distilled thoughts—intermediate reasoning results. This allows the model to retrieve and reuse high-level cognitive

patterns from past experiences efficiently, avoiding the noise and latency bottlenecks inherent in processing ultra-long raw contexts. In summary, unlike existing agent memory systems that store raw observations or skill descriptions, Thought-Retriever stores validated and deduplicated reasoning abstractions, offering a more information-dense and retrieval-friendly memory representation for agentic systems.

**Long-context LLMs.** In response to the challenge of long-context processing in LLMs, the most intuitive strategies involve expanding the LLM's context window. These methods include training larger, more advanced models (MosaicML, 2023; LM-SYS, 2023), fine-tuning existing language models to handle wider windows (Tay et al., 2022), applying positional encoding to extend the context window size (Xiao et al., 2023), and **compressing context via user embeddings (Ning et al., 2024)**. However, these methods often fall short due to the high costs associated with model training and a lack of flexibility. Moreover, simply extending the context does not guarantee faithful generation, as large models still suffer from issues like object hallucination, necessitating specific interventions (Li et al., 2025; 2024b), failing to fully address the fundamental reliability issues of long context.

**Retrieval-Augmented Language Models.** RALM offers a flexible, cost-effective alternative to long-context LLMs by retrieving relevant information—potentially assessed by LLMs themselves (Rahmani et al., 2024)—from context chunks. Current methods employ techniques like token embeddings (Izacard et al., 2022; Lin et al., 2023), keyword searches (Robertson et al., 2009), fine-tuned rerankers (Ram et al., 2023), and **parameterized retrieval verification losses (Fu et al., 2023)**. Recent paradigms, exemplified by **IRCoT** (Trivedi et al., 2023b) and emerging approaches like **RankCoT (Wu et al., 2025) and Rationale-Guided RAG (Sohn et al., 2025)**, attempt to enhance retrieval by interleaving or ranking step-by-step reasoning paths. However, these methods typically rely on *on-the-fly* generation of reasoning, which incurs prohibitive latency and computational overhead. Despite advancements like hierarchical tree structures (Chen et al., 2023) to manage context, existing solutions remain rigid or costly. We propose the Thought-Retriever framework using RALM, which efficiently condenses context into *stored* thoughts, addressing these efficiency challenges. To rigorously validate our approach against the evolving landscape of LLM capabilities (Pang et al., 2025) and evaluation standards (Li et al., 2024a; Kang et al., 2024), we also introduce a comprehensive benchmark.

**Context Compression for LLMs.** To alleviate context window constraints, context compression techniques aim to condense long inputs into compact representations. Early approaches focused on token-level pruning (Jiang et al., 2023). More recently, generative compression methods have gained traction. For instance, RECOMP (Xu et al., 2023b) and AutoCompressors (Chevalier et al., 2023) propose learning to synthesize compressive summaries or dense vectors to represent long documents. However, a significant drawback of these methods is the necessity to train additional compression modules or fine-tune the backbone LLMs, which incurs substantial computational overhead. Furthermore, methods relying on soft prompts or internal embeddings (Mu et al., 2023) are often incompatible with black-box APIs where model weights are inaccessible. In contrast, our Thought-Retriever is a lightweight, training-free framework. It is entirely model-agnostic, capable of seamlessly integrating with both locally deployed open-source models and closed-source commercial APIs, thus offering superior efficiency and broader real-world applicability while preserving high-level reasoning paths.

## 6 Conclusion

We introduce Thought-Retriever to enhance LLMs by dynamically generating and retrieving intermediate thoughts, enabling efficient use of external knowledge beyond context limits. For evaluation, we further propose AcademicEval, a benchmark for academic tasks like abstract and related work generation. Thought-Retriever outperforms existing methods, evolves through interaction, and shows strong potential for real-world applications. As a lightweight, model-agnostic memory module, Thought-Retriever can be readily integrated into LLM-based agentic systems to serve as their persistent, self-evolving long-term memory.

## 7 Limitations

Despite the promising results and contributions of our work, we would like to discuss some limitations. Our experiments and the AcademicEval dataset primarily utilize papers from AI-related fields, which could limit

the generalizability of our findings. Future work should consider extending the scope to a broader range of disciplines.

Additionally, our experiments and evaluations are conducted in English. This focus on English may overlook the nuances and challenges associated with other languages. Expanding our approach to include multilingual datasets and evaluations could provide a more comprehensive assessment of its effectiveness.

While AcademicEval provides a dynamic and continuously updated dataset from arXiv, it is reliant on the availability and quality of the papers uploaded to the platform. We assume and hope that researchers will continue to produce novel and high-quality work.

Lastly, while our framework shows effectiveness in our experiments, its robustness, scalability, and adaptability to real-world, extremely large-scale applications have yet to be fully tested. We are actively working on our demos and hope to provide more exciting updates on this front in the near future.

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

# A    Details of AcademicEval

In this section, we provide the **data format documentation** for the datasets in our proposed AcademicEval benchmark in Section A.1, and **detailed instructions and prompts** for its usage in Section A.2.

## A.1    Dataset Documentation

For AcademicEval-abstract, in the single document setting (Abstract-single), each case includes the paper title, abstract as the label, and main content, excluding the abstract and conclusion. For the multiple document setting (Abstract-multi), each case includes five papers' titles, abstracts, and main contents excluding the abstracts and conclusions. We utilize an expert LLM to summarize the five abstracts of one case into a fluent summary as its label using the prompt in Figure 11. For AcademicEval-related (Related-multi), each paper includes a title, its abstract, its related work as the label, the abstracts of its real citations, and the abstracts of other random papers.

|  | **Attribute** | **Description** |
|---|---|---|
| Abstract-Single | 'title' | The title of the academic paper. |
|  | 'abstract' as label | The abstract of the academic paper. |
|  | 'main_content' | The content of the paper excluding the abstract and the conclusion. |
| Abstract-Multi | 'title 1' | The title of the first academic paper. |
|  | 'abstract 1' | The abstract of the first academic paper. |
|  | 'main_content 1' | The content of the first paper excluding the abstract and the conclusion. |
|  | ... | ... |
|  | 'title 5' | The title of the fifth academic paper. |
|  | 'abstract 5' | The abstract of the fifth academic paper. |
|  | 'main_content 5' | The content of the fifth paper excluding the abstract and the conclusion. |
|  | 'label' | The summary of five abstracts as a fluent passage. |
| Related-Multi | 'title' | The title of the academic paper. |
|  | 'own abstract' | The abstract of the academic paper for wiring-related work. |
|  | 'own related work as label' | The related work of the academic paper for wiring-related work. |
|  | 'citations' abstracts' | The abstracts of the target paper's real citations. |
|  | 'other random abstracts' | The abstracts of other random papers. |

Table 4: **AcademicEval Dataset Documentation.** This table presents the specific format of the data in our AcademicEval dataset.

## A.2    Usage Instruction and Prompt Utilization.

Here we offer **detailed instructions** for utilizing the datasets in the AcademicEval benchmark. We also provide **all the necessary prompts we utilized in our experiment.**

**Abstract-Single.**   For the task of single paper abstract summarization, as shown in Figure 9 (a), we first provide a prompt *"Please craft an abstract summarizing the key points from the provided text. The abstract should be of appropriate length and include the main theme, significant findings or arguments, and conclusions of the text. Ensure it captures the essence of the content in a clear, succinct manner"* for the retrieval purpose. We then retrieve information from the paper's main content based on this prompt using a

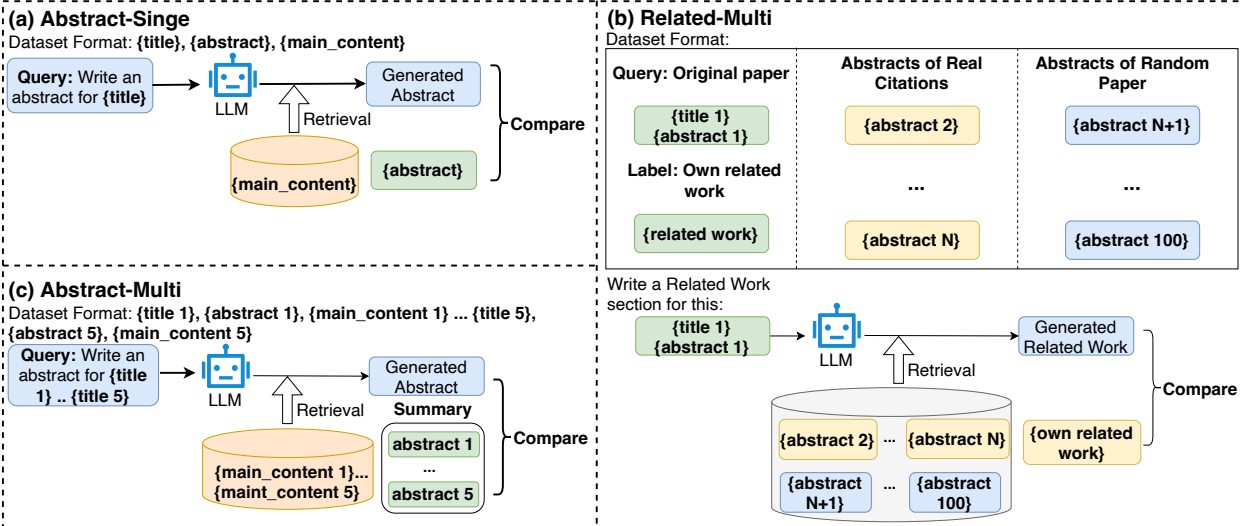

Figure 9: **AcademicEval Usage Instructions.** This figure provides a visualization of the usage instructions for the AcademicEval dataset, as described in Section A.2, to aid understanding.

retriever. Then, the LLM would generate an abstract based on the retrieved information using the prompt in Figure 10. Finally, we compare the LLM-generated abstract with the original abstract to do the evaluation.

**Abstract-Multi.** For the multiple paper abstracts summarization task, shown in Figure 9 (b), we first provide a prompt *"Please craft an abstract summarizing the key points from the provided text. The abstract should be of appropriate length and include the main theme, significant findings or arguments, and conclusions of the text. Ensure it captures the essence of the content in a clear, succinct manner"* for the retrieval purpose. Then we retrieve information from the main content of the 5 papers based on this prompt. Further, the LLM would generate an abstract based on the retrieved information with the prompt in Figure 10. The generated abstract is compared with the ground truth, which is a summary of the five abstracts created using the prompt in Figure 11.

**Related-Multi.** In the related work task, as shown in Figure 9 (c), we provide the LLM with a prompt *"Could you please write a related work for introducing this paper? Its abstract is: {paper_abs}"*, where *"{paper_abs}"* is sustibute with the paper's real abstract. Following this prompt, the LLM retrieves information from a collection of paper abstracts, comprising the abstracts of real citations in its related work section and random papers. The LLM then generates the related works section based on this retrieved information using the prompt in Figure 12. This generated related work is then compared with the real related work section of the paper to perform evaluation.

**Benefits and Contributions.** *AcademicEval* offers several advantages over existing benchmark datasets. *Firstly*, we maintain an up-to-date dataset from arXiv that benefits from the continuous publication of new papers. This dynamic nature eases overfitting and label leakage problems in static benchmarks and enables the evaluation of LLM self-adaptability. *Secondly*, high-quality labels can be generated with no extra cost as opposed to manually crafted datasets that require human effort. *Thirdly*, our dataset is not only valuable for evaluating LLM but also serves as a practical academic tool in the real world to assist researchers in better understanding their fields and boost productivity. We developed a highly automated codebase for dataset construction that will be released soon.

Please craft an abstract summarizing and connecting the key points from the provided Text.

The text should be composed of content extracted from different papers, potentially spanning varied disciplines, but all addressing overlapping themes or subjects."

The abstract should be of appropriate length (around 300 words), encompassing the main theme, significant findings or arguments, and conclusion of the Text.

Ensure the abstract captures the essence of the content in a clear, succinct manner, providing a coherent summary that bridges the various papers."
Here is the Text: {context}

Figure 10: **Prompt for Writing Abstracts.** This prompt was used in our experiment to ask the LLM to write an abstract based on the retrieved information. We provided in-context instructions to guide the LLM in producing higher-quality responses.

Create a concise, cohesive summary that encapsulates the key points and themes from the following five distinct abstracts. The summary should integrate the main ideas from each abstract to provide a comprehensive overview. It should be about 300 words.

Abstract 1: {abs1}

Abstract 2: {abs2}

Abstract 3: {abs3}

Abstract 4: {abs4}

Abstract 5: {abs5}

Figure 11: **Abstract Multi Ground Truth Prompt.** This prompt was used in our experiment on the Academic-abstract-multi dataset. Specifically, for each data entry, we summarize the abstracts of five papers in the entry to create the ground truth. To ensure high-quality generation, we utilized GPT-4o as the expert LLM to synthesize these summaries based on the provided prompt.

## B   Baseline Details

First, we consider 2 heuristic-based retrievers: (1) **BM25** (Robertson et al., 2009): A widely-used ranking function in information retrieval. (2) **TF-IDF** (Ramos et al., 2003): A statistical measure that evaluates the importance of a word in a memory.

Given the abstract and related work of a research article, along with a sample material, write a paragraph about its related work. Use the following as guidance:

Abstract: This research paper investigates the impact of climate change on global agricultural productivity. The study employs a comprehensive dataset of temperature and precipitation changes over the past century, combined with historical crop yield data. Through advanced statistical modeling and machine learning techniques, the research identifies significant correlations between temperature and precipitation fluctuations and variations in crop yields. Furthermore, it predicts future scenarios of agricultural productivity under different climate change scenarios, providing valuable insights for policymakers and stakeholders in the agricultural sector to develop adaptive strategies.

Related Work: Previous studies in the field have explored the relationship between climate change and agriculture but have primarily focused on specific regions or crops. Smith et al. (2017) conducted a comprehensive analysis of the impact of temperature on wheat yields in North America, highlighting the vulnerability of wheat crops to warming temperatures. Additionally, Johnson et al. (2019) investigated the effects of changing precipitation patterns on rice production in Southeast Asia, emphasizing the importance of water management in mitigating climate-related risks to agriculture. While these studies contribute valuable insights, our research extends their scope by considering a global perspective and employing advanced modeling techniques to provide more accurate predictions of future agricultural productivity under climate change scenarios.

Based on the abstract of this article and related materials, write a paragraph about its related work:
Abstract: {abstract}
Related materials: {context}

Figure 12: **Prompt for Writing Related Works.** This prompt was used in our experiment to ask the LLM to write a related work section based on the original paper's abstract and the retrieved related materials. We also provided an example of in-context learning to enable the LLM to perform more effectively on this challenging task.

Second, we select 4 deep learning-based retrievers: (3) **Contriever** (Izacard et al., 2022): leveraging contextualized embeddings and neural networks to understand and retrieve relevant memory chunks. (4) **DPR** (Karpukhin et al., 2020): retrieving memory chunks by encoding chunks and queries into dense vectors. (5) **DRAGON** (Lin et al., 2023): employing contrastive learning to train its ability to retrieve memory chunks. (6) **Qwen3-Embed-8b** (Zhang et al., 2025): A state-of-the-art decoder-only embedding model

Template-based Query Formation:
What new perspectives does '{title}' offer in its field
How might the findings in '{title}' influence future research?
What are the practical applications of the research in '{title}'?
In what ways does '{title}' challenge existing theories or beliefs?
How does '{title}' contribute to our understanding of its subject matter?
What does the statement '{sentence}' imply in the context of '{title}'?
How does the sentence '{sentence}' relate to the overall theme of '{title}'?

LLM-based Query Formation:
Given the paper title: {title}; and its abstract {abstract}, please ask 20 questions that would be helpful for writing its related work section. Each questions should have a number at the begining. For example:\n 1.<Put Your Question Here>\n2. <Put Your Question Here>, etc. The questions should be diverse and with different level of abstraction.

Figure 13: **User Query Formation Prompt.** This figure presents the prompt used to model real-world user queries. Specifically, it includes two methods: template-based query formation, where general question templates are created to be suitable for a wide range of papers, and LLM-based query formation, where this prompt is used to ask an LLM to generate diverse queries.

Given the original abstract:{original},and given the two generated abstracts:

Generated Abstract 1:{gen1};  and Generated Abstract 2: {gen2}, plase evaluate which one is closer to the original abstract.

Just output 'Abstract 1 is better' or 'Abstract 2 is better', no extra words.

Figure 14: **AI Evaluation Prompt.** This prompt is used for the AI Evaluation metric Win Rate, as described in Section 4. Given two generated answers and the ground truth answer, we ask the expert LLM to determine which generated answer aligns more closely with the ground truth.

that leverages large-scale pre-training to achieve superior semantic understanding and retrieval performance across diverse tasks.

Third, we include 2 advanced retrieval and compression strategies: (7) **IRCoT** (Trivedi et al., 2023a): An iterative framework that interleaves retrieval with Chain-of-Thought reasoning, allowing the model to dynamically retrieve information based on partial reasoning steps for complex queries. (8) **RECOMP** (Xu et al., 2023a): A retrieval augmentation method that compresses retrieved documents into concise summaries or selects key segments to maximize information density within the context window.

Fourth, we consider full context window baselines with document truncation: (9) **Full Context (left)** (Chen et al., 2023): This approach uses the initial segment of a document, truncated to fit within a 4,096-token window. Focusing on the first 4,096 tokens, it prioritizes early content in the document. (10) **Full Context (right)** (Chen et al., 2023): In contrast to Full Context (left), it utilizes the final segment of a document, also truncated to a 4,096-token window.

Lastly, we selected two long-context LLMs: (11) **OpenOrca-8k** (Mukherjee et al., 2023): is fine-tuned on the Mistral 7B model using the OpenOrca dataset. At its release time, it was ranked the best model among all models smaller than 30B on Hugging Face, with a maximum context length of 8,192 tokens. (12) **Nous Hermes-32k** (Shen et al., 2023): trained on Mixtral8x7B MoE LLM. It boasts a maximum context length of 32,768 tokens.

Note that we do not compare with MEMWALKER (Chen et al., 2023), since it is costly to run and cannot scale to tasks with many data chunks. We use Contriever as Thought-Retriever's retriever.

## C  Retrieve Context Generated from other LLMs

We utilize an example shown in Figure 4 to illustrate our LLM interacting with four other specialized LLMs, each an expert in a distinct field. These expert LLMs assume designated roles (such as a doctor) and possess unique background knowledge. Consequently, our LLM can quickly assimilate its insights and integrate this external knowledge.

## D  User Query Formation

To model user queries in real-world scenarios for guiding thought generation, we primarily use two approaches: 1) template-based query formation, and 2) LLM-based query formation. The prompts are shown in Figure 13

**Template-based Query Formation.**    We construct general and broadly applicable templates for all papers. For example, *"What are the practical applications of the research in 'title'?"* and *"What new perspectives does 'title' offer in its field?".* During experiments, we substitute *'title'* with the actual paper title to form specific queries.

**LLM-based Query Formation.**    Another approach we use to generate more specific queries is by leveraging LLMs. Specifically, we utilize models such as Mistral 8x7B and expert LLM. By providing these models with the paper title and abstract, we ask them to generate diverse questions at varying levels of abstraction. These questions are tailored to each specific paper, allowing for more nuanced and targeted queries.

## E  Specific Queries of Abstract Level

This section lists the specific queries utilized in our case study in Section 4.4, demonstrating how Thought-Retriever leverages deeper thoughts for more abstract user queries. Each query is categorized by its general level of abstraction, ranked according to its abstraction level as assessed by an expert LLM, and detailed with its exact content in Table 6.

### E.1  Algorithm Adaptability and Filter Effectiveness

**Thought-Retriever is adaptable to various LLM backbones.**    While we use carefully designed prompt templates, Thought-Retriever is not tailored to any specific model. The algorithm is adaptable and effective across various LLM backbones, as shown by the consistent top performance on both Qwen-7B Bai et al. (2023) and Llama-3-70B Dubey et al. (2024) in multiple tasks (Table 7).

## F  Example Outputs Comparison of Different Methods

We present examples of outputs generated using different methods on the AcademicEval-abstract-single dataset. Specifically, in Figure 15, we provide the original paper title and abstract, along with the abstract generated by our Thought-Retriever, **accompanied by a comment from an expert LLM**. In Figure 16, we show abstracts generated using DPR and TF-IDF, also **accompanied by expert LLM comments** for

Table 5: **Example of thought construction**. For a given query, Thought-Retriever retrieves the corresponding data chunks (thoughts and original data chunks) from the knowledge corpus to get the response. Then, Thought-Retriever integrates the query and response into a prompt through the prompt template and obtains the final thought candidate through the output of LLM. This thought candidate is then evaluated to determine whether it is correct and not redundant.

| | |
|---|---|
| Query | What has driven significant progress in various NLP tasks in recent years? |
| Response | According to the text, significant progress in various NLP tasks in recent years has been driven in part by benchmarks such as GLUE, whose leaderboards rank models by how well they perform on these diverse tasks. |
| Prompt template | Input: Given query:{query}, given response:{response}. Based on the provided query and its corresponding response, perform the following step: succinctly summarize both the question and answer into a coherent knowledge point, forming a fluent passage. |
| Thought candidate | Here is a summarized knowledge point: In recent years, significant progress has been made in various Natural Language Processing (NLP) tasks. A key driver of this progress is the development of benchmarks, such as GLUE, which provide a standardized way to evaluate and compare the performance of different models on a range of diverse NLP tasks. These benchmarks, which often take the form of leaderboards, rank models based on their performance, fostering competition and innovation in the field. As a result, researchers and developers have been motivated to improve their models, leading to significant advancements in NLP capabilities. |

| Abstraction | Rank (expert LLM) | Query |
|---|---|---|
| High | 6 (Most Abstract) | "What are the broader future implications of user-centric utility in NLP model evaluation?" |
| High | 5 | "Please craft an abstract summarizing the key points from the provided text." |
| Medium | 4 | "What are some of the limitations of this study?" |
| Medium | 3 | "What are the key methods introduced in this paper?" |
| Low | 2 | "Please explain the term Minerva to me." |
| Low | 1 (Least Abstract) | "How many benchmarks are used to test the model's long context understanding ability in this paper?" |

Table 6: **Sample Queries Used in Abstraction Level Case Study.** This table presents sample queries from the case study conducted in Section 4, which demonstrates how Thought-Retriever learns to leverage deeper thoughts to answer more abstract user queries.

comparison. In Figure 17, we showed the example abstract generated by the long context model Nous Hermes 32k and the **corresponding comments from the expert LLM.** It is evident that the abstract generated by our Thought-Retriever is more comprehensive and coherent, with better management of specification and abstraction levels. Below, we include a comprehensive comment from the expert LLM:

*"**The Thought-Retriever abstract is the best and most aligned with the original abstract.** It effectively **captures all key points**, including the critique of leaderboard metrics and the need to consider factors beyond accuracy, such as energy efficiency, model size, and inference latency. It also calls for increased transparency on leaderboards, emphasizing a holistic approach to NLP evaluation that includes practical statistics*

Table 7: **Thought-Retriever demonstrates adaptability across different LLMs**. This table compares Thought-Retriever's performance against baselines on Abstract-single and Abstract-multi tasks using Qwen-7B and Llama-3-70B models. Thought-Retriever consistently delivers the best results, highlighting its adaptability to various LLMs.

| Type | Abstract-single | | | | Abstract-multi | | | |
|------|-----------------|--|--|--|----------------|--|--|--|
| LLM | Qwen-7b | | Llama-3-70b | | Qwen-7b | | Llama-3-70b | |
| Method | F1 | Win Rate | F1 | Win Rate | F1 | Win Rate | F1 | Win Rate |
| BM25 | 0.196 | 3% | 0.22 | 13% | 0.232 | 7% | 0.233 | 14% |
| TF-IDF | 0.192 | 3% | 0.21 | 17% | 0.220 | 3% | 0.224 | 12% |
| Contriever | 0.231 | 10% | 0.238 | 18% | 0.229 | 4% | 0.228 | 19% |
| DPR | 0.209 | 4% | 0.215 | 18% | 0.222 | 3% | 0.222 | 11% |
| DRAGON | 0.209 | 3% | 0.225 | 13% | 0.224 | 4% | 0.236 | 19% |
| Full Context (left) | 0.069 | 0% | 0.102 | 0% | 0.061 | 0% | 0.107 | 0% |
| Full Context (right) | 0.073 | 0% | 0.104 | 0% | 0.065 | 0% | 0.103 | 0% |
| OpenOrca-8k | 0.175 | 17% | 0.175 | 23% | 0.135 | 17% | 0.135 | 10% |
| Nous Hermes-32k | 0.247 | 20% | 0.247 | 37% | 0.204 | 13% | 0.204 | 7% |
| **Thought-Retriever** | **0.259** | **50%** | **0.285** | **50%** | **0.253** | **50%** | **0.266** | **50%** |

*to provide a comprehensive measure of model utility. This abstract is **clear, well-organized, and includes a call to action** for changes in leaderboard reporting to better serve the practical needs of NLP practitioners."*

*"In contrast, the **DPR and long context model abstract, while touching on similar points, is less comprehensive and focuses more on specific suggestions** like user-specific leaderboards and revealed preference theory without fully encapsulating the broader argument about the divergence between leaderboard metrics and practitioner needs. The **TFIDF abstract diverges the most**, discussing related topics like brittleness, bias, and out-of-distribution data, but it **does not focus specifically on the central argument** about leaderboard metrics versus practical utility, making it less aligned with the original abstract's intent."*

## G   Discussion

**Transformative Impact and Real-World Applications.** The Thought-Retriever represents a paradigm shift in AI systems, transforming them from static repositories of knowledge to dynamic, intelligent frameworks that interact and learn. Its unique architecture not only processes and retrieves information but also evolves with each user interaction, effectively 'thinking' and adapting over time. Such an intelligent system is crucial for scenarios where real-time learning and context-aware responses are vital. For instance, existing AI service systems could be significantly enhanced by incorporating our approach. By storing original guidelines and regulations as part of the external knowledge base and recording each human query and its results as thoughts, these systems can evolve into more intelligent entities capable of continuous improvement and learning. This adaptive capability makes the Thought-Retriever an invaluable tool for dynamic and ever-changing industrial environments, where quick decision-making based on historical data and evolving information is crucial. In sectors like customer service, healthcare, and legal advisory, where personalized and informed responses are key, the Thought-Retriever can provide more accurate, context-aware, and efficient solutions. Its ability to continuously learn and adapt from user interactions positions it as a groundbreaking tool for transforming how industries interact with and utilize AI technology.

**Future Research.** Inspired by human thinking, our Thought-Retriever represents a solid step toward general AI agents. Building on this foundation, future research could address several key challenges. Firstly, scalability and efficiency in processing increasingly complex datasets will be crucial. This involves not only enhancing computational power but also refining algorithms for greater precision and speed. Secondly, understanding and mimicking human-like reasoning remains a pivotal goal. This includes grasping nuances in language, emotion, and cultural contexts, and pushing the boundaries of what AI can comprehend and respond to. Moreover, ensuring ethical considerations in AI decision-making is significant. As the retriever evolves, its impact on privacy, security, and societal norms must be rigorously evaluated and guided. Finally,

explore new domains of application, such as personalized education, mental health analysis, and advanced robotics.

## H  Arxiv Copilot Demo

Based on the Thought-Retriever, we further propose a demo named Arxiv Copilot and deploy it on the Huggingface shown in Figure 18, which aims to provide personalized academic service. More specifically, it consists of three main parts as follows. Firstly, in the first "Profile" part, users can enter the researcher's name and generate a research profile. Secondly, in the research trend part, users can select a time range and get relevant topic trends and ideas. Finally, in the "Chat and Feedback" part, users can chat with Arxiv Copilot and choose the better response from two answers. Here, we appreciate any further feedback.

**Profile**   In this part, as shown in Figure 18 (a), the user can input his/her name in a standard format to get the profile from arXiv here.

**Research Trend**   As shown in Figure 18 (b), Arxiv Copilot will give the user personalized research trends and ideas if the user has set his/her profile. Otherwise, general research trends will be provided.

**Chat and Feedback**   As shown in Figure 18 (c), each time Arxiv Copilot will give two answers. If the user prefers the second answer, he/she can click 'like' below the second answer, and the first answer will be removed. If the user clicks 'dislike', the second answer will be removed.

## I  Further Results on QA and Reasoning Task

We evaluated Thought-Retriever on the recent LooGLE dataset Li et al. (2023) for QA and reasoning tasks. As shown in the Table 8, it consistently outperformed all baselines, demonstrating strong performance in both QA and reasoning accuracy.

Table 8: **Thought-Retriever's Effectiveness on QA and Reasoning Tasks.** This table presents results from LooGLE, a recent and widely used QA and reasoning benchmark. Our Thought Retriever consistently outperforms all other baselines.

|  | QA Accuracy | Reasoning Accuracy |
|---|---|---|
| BM25 | 10% | 30% |
| TF-IDF | 13% | 33% |
| Contriever | 20% | 50% |
| DPR | 17% | 20% |
| Dragon | 13% | 27% |
| Full Context (left) | 7% | 20% |
| Full Context (right) | 7% | 13% |
| OpenOrca - 8K | 0% | 10% |
| Nous Hermes-32K | 3% | 17% |
| Thought Retriever | **27%** | **57%** |

## J  API Acknowledgement

We used Together AI's API to conduct our experiments. There are no specific requirements to run our code. Essentially, our experimental setup can be replicated by anyone with standard laptops or desktop computers and any compatible API, not necessarily Together AI's API.

Table 9: **Ablation with latest retriever models.** We substitute the Contriever component in Thought-Retriever with NV-Embed, one of the latest retriever models on MTEB, and evaluate their performance comparatively on the Abstract Single task.

| Method | F1 |
|---|---|
| Contriever | 0.242 |
| Thought-Retriever using Contriever | 0.290 |
| NV-Embed | 0.268 |
| Thought-Retriever using NV-Embed | 0.326 |

Table 10: **Comparison between Thought-Retriever and RECOMP in Abstract-single and Abstract-multi.**

| Dataset | Model | F1 | Win Rate |
|---|---|---|---|
| Abstract-single | RECOMP | 0.237 | 7% |
| Abstract-single | Thought-Retriever | 0.290 | 50% |
| Abstract-multi | RECOMP | 0.202 | 8% |
| Abstract-multi | Thought-Retriever | 0.275 | 50% |

## K   Diversity of user queries and thoughts in real-world applications

We discuss the diversity of user queries and thoughts in real-world applications from the following three perspectives.

**Redundancy Filtering for Diversity.**   As stated in section 2.3, in our framework, when a new thought is generated, we explicitly check its cosine similarity against existing thoughts in the pool. If it is too similar to any existing thought, it is discarded. This ensures that unique, non-redundant thoughts are retained, maintaining the diversity of the thought pool.

**Empirical Diversity Analysis.**   To quantify diversity, we further conducted an analysis where we randomly sampled 50 thoughts from the thought pool in the AcademicEval task and computed their pairwise cosine similarities. This was repeated 10 times. The average pairwise similarity was 0.32, indicating a high level of semantic diversity among the thoughts.

**Preliminary Real-World Usage Data.**   As introduced in Appendix H, we have built an Arxiv Copilot system based on the Thought-Retriever, and it is capable of interacting with real users. From actual usage data collected through our deployed system, we observe a wide variety of user query types and thought interactions. This real-world evidence further supports the claim that our system encourages and handles diverse, meaningful memory retrieval in practical scenarios.

## L   Comparison between Thought-Retriever and other SOTA baselines

**Ablation with latest retriever models.**   We conducted an experiment on the Abstract Single task using the latest retriever models on MTEB - NV Embed Lee et al. (2024), and the results are shown in Table 9. These results further support that our framework and retrievers are mutually reinforcing. On a side note, many of the latest retriever models on MTEB, including NV-Embed (7.85B parameters), are substantially larger than Contriever (110M parameters) and require significantly more computational resources. For our initial experiments, we prioritized models that balance performance with computational efficiency to ensure broader accessibility and practical implementation of our framework.

**Comparison with newest long-context understanding models.**   We introduced an advanced retrieval algorithm named RECOMP (Xu et al., 2023b) for comparison. RECOMP is a strong baseline that tackles the

long-context issue by combining an extractive compressor, which selects key sentences from documents, with an abstractive compressor, which creates summaries by synthesizing information across multiple documents. We compared Thought-Retriever with RECOMP on the Abstract-single and Abstract-multi tasks. The results, presented in Table 10, further validate the effectiveness of our method in understanding long-context.

## M    Details of our thought filtering mechanisms

We discuss our thought filtering mechanisms from the following three aspects.

**Thought Quality Filtering.**   When generating answers based on a given query and retrieved information, we explicitly instruct the LLM to assess whether the retrieved content is sufficient to answer the question. If not, it is prompted to return `"No"` rather than produce a speculative answer. Furthermore, when generating thoughts based on the question and answer, we use a specialized prompt that includes a binary confidence check—asking the model to indicate whether the generated thought is logical and non-hallucinated. If not meaningful, the thought is discarded. This process is detailed in section 2.3, and the full prompt is included in Figure 3.

**Thought Redundancy Filtering.**   As illustrated in section 2.3, before adding a new thought to the pool, we compute its cosine similarity against existing thoughts. If it is too similar to any existing entry, it is filtered out. This ensures that unique, non-redundant thoughts are stored, promoting diversity and reducing duplication.

**Robustness to Low-Quality Thoughts − Added Empirical Study.**   We further add an experiment that shows that our Thought-Retriever is robust even when some low-quality or distracting thoughts are present. Specifically, we designed an experiment using the *Related-multi* dataset, where the ground truth of retrieval is known (i.e., the abstract chunks of the real citation paper). We compared two methods to assess the impact of low-quality thoughts:

- **Without relevant thoughts** – the retrieval process does not include the ground-truth abstract chunks, and irrelevant thoughts are added.

- **Raw data chunks only** – no thoughts are added at all.

The F1 performance of the two methods (**Without relevant thoughts**: 0.207; **Raw data chunks only**: 0.208) demonstrates that the performance of the Thought-Retriever is not significantly affected by extremely low-quality thoughts.

## N    Computational analysis

In the RAG system, the majority of computation is allocated to LLM inference, while retrieval operations can be executed on the CPU. The Thought-Retriever maintains nearly identical retrieval compute costs across all experiments by using the same retriever. The cost of LLM inference is directly proportional to the number of retrieved tokens. To illustrate the efficiency of Thought-Retriever in real-world applications, we use the Arxiv Copilot introduced in Appendix H as an example. We have constructed 100,000 paper abstracts as the original data chunks, and we pre-calculate and store the embeddings of these chunks. Additionally, we need to construct index files to establish the connection between data chunks and thoughts. The storage of this data requires less than 1.5 GB of memory. On this basis, we utilize FAISS[2], a high-efficiency vector processor capable of handling billions of vectors, as the similarity retriever, and implement the entire framework on CPUs. When a user query arrives, we use the Thought-Retriever based on FAISS to retrieve historical thoughts or data chunks and save the newly derived thoughts. The average inference time for generating user responses is approximately 5 seconds, which is mainly limited by the API's response speed.

---

[2]https://github.com/facebookresearch/faiss

Table 11: Human evaluation instructions for thought quality assessment.

---

**Task Description:**
You will be shown a query and several intermediate thoughts generated by a large language model (LLM). Please evaluate each thought **independently** based on its usefulness, relevance, and correctness. Your rating should reflect how well the thought contributes to answering the query.

**Scoring Criteria (per thought):**
  **Score 2 (High Quality)**:     Relevant, helpful, logically coherent, and factually correct.
  **Score 1 (Medium Quality)**:   Somewhat relevant/helpful; may have minor issues or vague content.
  **Score 0 (Low Quality)**:     Irrelevant, incorrect, incoherent, or hallucinated content.

**Instructions:**

- Evaluate each thought **individually**, without comparing it to others.

- Focus on content quality, not writing style or fluency.

- You only need to assign a score (0, 1, or 2); no explanation required.

**Example:**
**Query:** What are the key challenges in training large-scale language models?
**Thought A:** One major challenge is the need for large-scale, high-quality datasets that accurately reflect diverse linguistic contexts.
**Thought B:** Transformer models were introduced in 2017 by Vaswani et al., and they changed NLP.
**Thought C:** LLMs are expensive to train and often face issues like overfitting and gradient instability.
**Recommended Scores:**
Thought A $\rightarrow$ 2     Thought B $\rightarrow$ 0     Thought C $\rightarrow$ 2

---

## O   Human evaluations on the quality of retrieved thoughts

We conduct human evaluations on the quality of retrieved thoughts from the following three aspects.

**Human Evaluation of Thought Quality.** We conducted a human evaluation study (the detailed human instructions can be seen in Table 11) to directly assess the quality and reliability of the generated thoughts and our confidence-based filtering mechanism. Ten volunteers were recruited to review generated thoughts, with each thought independently evaluated by five annotators. On the **Abstract-single** dataset, the thoughts matched human judgment 96% of the time. On the **Related-multi** dataset, the agreement rate was 93%. These results confirm that our LLM-generated confidence scores are highly reliable and effective for filtering out low-quality or hallucinated thoughts.

**Robustness to Low-Quality Thoughts (Empirical Study).** We further evaluated how the system performs in the presence of low-quality thoughts. Using the **Related-multi** dataset, we compared two settings: (a) **Without relevant thoughts**, where only unrelated thoughts were injected, and (b) **Raw data chunks only**, without any thoughts. The F1 scores in both settings were nearly identical (0.207 vs. 0.208), indicating that our framework remains robust even in the presence of noisy or irrelevant thoughts.

**Real-World Deployment Validation.** As introduced in Appendix H, we have built a real-world application named *Arxiv Copilot* with Thought-Retriever. Arxiv Copilot can interact with real users and provides two types of answers: one based solely on retrieved original data chunks, and the other incorporating both original data chunks and historical thoughts. We collected feedback from 500 real users to compare preferences for these two types of responses. Approximately 75% of users preferred answers that included both original data chunks and historical thoughts. This observation demonstrates the effectiveness of Thought-Retriever in real-world scenarios involving large query requests.

# P   Comparison with other baselines

## P.1   Comparison with MemWalker

In this section, we compare Thought-Retriever with MemWalker (Chen et al., 2023), a representative method designed for processing ultra-long contexts through interactive reading.

**MemWalker Setting.** MemWalker operates by first segmenting the long context and constructing a hierarchical tree of summaries. During the inference phase, it treats the retrieval process as an interactive decision-making task. The LLM acts as an agent that "walks" down this memory tree: at each step, the model reads the current summary node, reasons about which child node to visit next, and iteratively navigates until it locates the relevant leaf nodes. For our baseline implementation, we utilized the standard configuration with a branching factor of 3 and a maximum depth of 3 as proposed in (Chen et al., 2023). **We justify this configuration based on three factors:** 1) *Coverage:* A depth of 3 yields a capacity of $3^3 = 27$ leaf nodes (covering ∼13,500 tokens), which aligns well with the average length of academic papers in our dataset; 2) *Structure:* The hierarchy mimics the natural "Title → Section → Content" structure of academic texts; 3) *Context Window:* A branching factor of 3 ensures that the parent summary context at each decision step remains small enough to fit within the LLM's effective window, preventing navigation errors caused by information overload.

**Performance and Efficiency Analysis.** Table 12 presents the detailed comparison. Thought-Retriever consistently outperforms MemWalker in both F1 score and Win Rate across all datasets (e.g., 0.290 vs. 0.268 on *Abstract-single*). More critically, in terms of inference time, MemWalker suffers from significant latency due to its sequential nature; each query requires multiple rounds of LLM inference to navigate the tree structure. For instance, on the *Abstract-multi* task, MemWalker takes 85.0 seconds per query. In contrast, Thought-Retriever completes the same task in just 3.20 seconds—achieving a speedup of over 25×. This demonstrates that while MemWalker is effective, Thought-Retriever offers a superior balance of accuracy and efficiency for real-world applications.

Table 12: Performance and Inference Time Comparison across Datasets. Note that MemWalker incurs high latency due to its interactive, multi-step tree navigation process.

| Type | Dataset | MemWalker | | | Thought-Retriever (Ours) | | |
|------|---------|-----|----------|------|-----|----------|------|
| | | F1 | Win Rate | Time | F1 | Win Rate | Time |
| Academic Eval | Abs-single | 0.268 | 40% | 25.0s | **0.290** | **50%** | **1.85s** |
| | Abs-multi | 0.255 | 35% | 85.0s | **0.275** | **50%** | **3.20s** |
| | Rel-multi | 0.212 | 44% | 65.0s | **0.216** | **50%** | **2.40s** |
| Public | Gov Report | 0.229 | 45% | 30.0s | **0.232** | **50%** | **3.50s** |
| | WCEP | 0.228 | 39% | 28.0s | **0.238** | **50%** | **3.10s** |

## P.2   Comparison with Oracle

To rigorously evaluate the effectiveness of our proposed method, we introduce an **Oracle** baseline which serves as a theoretical upper bound. Specifically, we utilize the ground truth answers to calculate the ROUGE-L overlap scores with all corpus chunks, selecting the top-$K$ chunks with the highest overlap to serve as the "perfect" input context. **Crucially, we employ the same generator (Mistral-8x7B) and context window constraints** for the Oracle, Thought-Retriever, and all baselines. This ensures that any performance difference is solely attributed to the quality and format of the retrieved context.

Table 13 reveals a counter-intuitive yet compelling result: **Thought-Retriever consistently outperforms the Oracle baseline on AcademicEval datasets** (e.g., F1 0.290 vs. 0.278 on Abstract-single), while maintaining a competitive >50% win rate against traditional baselines. Although the Oracle utilizes ground-truth-derived gold chunks, these raw texts inherently contain significant redundancy and irrelevant syntactic

Table 13: **Performance comparison with the SOTA baseline and the Oracle upper bound.** We select Qwen3-Embed-8b as the strongest baseline. Win rate compares each method's response with Thought-Retriever (50% represents a tie). **Remarkably, Thought-Retriever achieves a >50% win rate against the Oracle on AcademicEval tasks**, demonstrating that generated thoughts provide superior reasoning guidance than raw gold chunks.

| Method | Abstract-single | | Abstract-multi | | Related-multi | | Gov Report | | WCEP | |
|---|---|---|---|---|---|---|---|---|---|---|
| | F1 | Win Rate | F1 | Win Rate | F1 | Win Rate | F1 | Win Rate | F1 | Win Rate |
| SOTA Baseline (Qwen3) | 0.245 | 28% | 0.240 | 20% | 0.211 | 35% | 0.229 | 42% | 0.235 | 44% |
| *Oracle (Gold Chunks)* | *0.278* | *46%* | *0.255* | *42%* | *0.214* | *49%* | *0.255* | *62%* | *0.245* | *55%* |
| **Thought-Retriever** | **0.290** | **50%** | **0.275** | **50%** | **0.216** | **50%** | **0.232** | **50%** | **0.238** | **50%** |

noise. Given the fixed context window (e.g., 2,000 tokens), filling the prompt with raw chunks limits the total volume of distinct information the model can ingest.

In contrast, our method generates *thoughts* that act as highly compressed, information-dense representations of the retrieved content. This compression allows Thought-Retriever to "pack" a broader scope of relevant details into the same context limit compared to the verbose raw text used by the Oracle. Consequently, in complex multi-hop scenarios (AcademicEval) where information coverage is paramount, our method provides the generator with a more comprehensive evidence set than even the gold raw chunks. While the Oracle regains a slight lead in pure extraction tasks (Public datasets), Thought-Retriever significantly narrows the gap compared to the strongest baseline (Qwen3-Embed-8b), demonstrating the robustness of thought-based retrieval in maximizing information density.

### P.3 Retrieval Granularity and Chunk Coverage Analysis against HRALM

To rigorously evaluate the retrieval granularity, we assess the quality of the retrieved context by measuring its coverage of the ground truth evidence. Since both HRALM (we use MemWalker (Chen et al., 2023) here) and Thought-Retriever retrieve higher-level abstractions (summaries $S$ or thoughts $T$) rather than raw text, simply counting retrieved items is insufficient. As illustrated in Figure 1 and defined in Section 2.1, we utilize the root source mapping function $\hat{O}(\cdot)$ to trace retrieved abstractions back to their constituent raw data chunks $\mathcal{K}_{retrieved}$. We then compare these mapped chunks against the Gold Chunks set $\mathcal{K}_{gold}$ (identified by the Oracle as having the highest ROUGE-L overlap with the ground truth answer).

- *Precision:* The proportion of mapped raw chunks that are relevant: $|\mathcal{K}_{retrieved} \cap \mathcal{K}_{gold}|/|\mathcal{K}_{retrieved}|$.

- *Recall:* The proportion of gold chunks covered by the retrieval: $|\mathcal{K}_{retrieved} \cap \mathcal{K}_{gold}|/|\mathcal{K}_{gold}|$.

As shown in Table 14, determining the root source reveals a structural trade-off between the two methods. In focused tasks like Abstract-single, HRALM achieves slightly higher precision (0.82 vs. 0.80) because its summarization process creates clean boundaries around specific semantic clusters, introducing less noise when the query falls neatly into one cluster. However, its rigid hierarchy struggles with queries requiring cross-cluster information, leading to low recall in Rel-multi (0.52). In contrast, Thought-Retriever leverages thoughts to bridge semantic gaps by retrieving content that inherently connects disparate raw chunks. This capability allows it to achieve significantly higher recall (0.82) and F1 (0.79), demonstrating superior coverage for complex reasoning tasks.

## Q   Causal Analysis of Self-Evolution on Held-out Sets

To verify that the performance improvement is causally linked to the accumulation of thoughts rather than mere correlation, we conducted a rigorous held-out evaluation on both *Abstract-multi* and *Related-multi* datasets.

**Experimental Setup.** We partition each dataset into two disjoint subsets with a **50%/50% split ratio**:

Table 14: Retrieval Quality Comparison: HRALM vs. Thought-Retriever. We evaluate the **Chunk Coverage** performance based on the precision and recall of retrieved raw data chunks. **Note:** While HRALM achieves slightly higher precision in single-document tasks (e.g., Gov Report) due to its extraction-based summarization, Thought-Retriever significantly outperforms it in recall and overall F1 score by bridging semantic gaps with generated thoughts.

| Type | Dataset | HRALM (Baselines) | | | Thought-Retriever (Ours) | | |
|---|---|---|---|---|---|---|---|
| | | Prec. | Recall | F1 | Prec. | Recall | F1 |
| AcademicEval | Abs-single | **0.82** | 0.65 | 0.72 | 0.80 | **0.88** | **0.84** |
| | Abs-multi | 0.74 | 0.58 | 0.65 | **0.78** | **0.84** | **0.81** |
| | Rel-multi | 0.65 | 0.52 | 0.58 | **0.76** | **0.82** | **0.79** |
| Public | Gov Report | **0.78** | 0.60 | 0.68 | 0.75 | **0.82** | **0.78** |
| | WCEP | 0.71 | 0.62 | 0.66 | **0.77** | **0.83** | **0.80** |

- **Evolution Set ($Q_{evol}$):** The first 50% of queries are used solely for generating and accumulating thoughts into the memory $\mathcal{T}$. No performance metrics are recorded during this phase.

- **Held-out Test Set ($Q_{test}$):** The remaining 50% of queries are used for evaluation. These queries are strictly unseen during the evolution phase.

**Results.** We compare the F1 performance on the *Test Set* under two experimental conditions:

1. **Baseline (Cold Start):** The model processes the test queries starting with an empty thought memory ($\mathcal{T} = \emptyset$).

2. **Ours (Evolved):** The model processes the test queries with a thought memory pre-filled by the Evolution Set ($\mathcal{T} \leftarrow$ Thoughts from $Q_{evol}$).

As presented in Table 15, the model utilizing the evolved memory achieves consistent performance gains across both datasets. Specifically, on *Abstract-multi*, the pre-accumulated thoughts yield a **6.4%** relative improvement in F1 score compared to the cold-start baseline. This provides strong causal evidence that the system learns transferable reasoning patterns from past interactions that generalize to novel, unseen queries.

Table 15: Causal Evidence of Self-Evolution on Disjoint Sets. Comparison of F1 scores on the held-out Test Set (50% of data) with and without prior evolution on the disjoint Evolution Set (50% of data). The consistent improvement confirms that accumulated thoughts provide transferable benefits to novel queries.

| Method | Memory Status (at start of testing) | F1 Score on Held-out Test Set | |
|---|---|---|---|
| | | Abstract-multi | Related-multi |
| **Baseline (Cold Start)** | Empty | 0.235 | 0.198 |
| **Ours (Evolved)** | Pre-filled (from Evolution Set) | **0.250** | **0.210** |
| | *Relative Improvement* | **+6.4%** | **+6.1%** |

## R  Inference Time Comparison

In this section, we analyze the computational efficiency of Thought-Retriever compared to baseline methods. We measured the average end-to-end inference latency per query on a single NVIDIA A6000 GPU. The results are summarized in Table 16.

Table 16: **Inference time comparison across different datasets.** Values represent the average latency in seconds (s) per query. Lower is better. While heuristic and simple dense retrievers are the fastest, **Thought-Retriever** maintains a competitive inference speed compared to advanced baselines like IRCoT and long-context LLMs (Nous Hermes-32k), achieving a balance between performance and efficiency.

| Type | Inference Time (Seconds) ↓ | | | | |
|---|---|---|---|---|---|
| **Dataset** | **Abstract-single** | **Abstract-multi** | **Related-multi** | **Gov Report** | **WCEP** |
| Method | Time (s) | Time (s) | Time (s) | Time (s) | Time (s) |
| BM25 | 0.42 | 0.45 | 0.48 | 0.75 | 0.62 |
| TF-IDF | 0.40 | 0.43 | 0.46 | 0.72 | 0.60 |
| Contriever | 0.65 | 0.72 | 0.68 | 1.10 | 0.95 |
| DPR | 0.62 | 0.70 | 0.66 | 1.05 | 0.92 |
| DRAGON | 0.68 | 0.75 | 0.70 | 1.12 | 0.98 |
| Qwen3-Embed-8b | 2.15 | 2.30 | 2.25 | 2.85 | 2.60 |
| IRCoT | 4.80 | 10.50 | 6.20 | 7.50 | 6.80 |
| RECOMP | 1.25 | 1.55 | 1.40 | 2.50 | 2.10 |
| Full Context (left) | 2.80 | 3.10 | 3.05 | 5.20 | 4.10 |
| Full Context (right) | 2.80 | 3.10 | 3.05 | 5.20 | 4.10 |
| OpenOrca-8k | 4.50 | 5.20 | 5.10 | 9.80 | 7.50 |
| Nous Hermes-32k | 12.50 | 14.20 | 13.80 | 28.50 | 19.20 |
| **Thought-Retriever** | **1.85** | **3.20** | **2.40** | **3.50** | **3.10** |

**Comparison with Heuristic and Dense Retrievers.** As expected, lightweight heuristic methods (e.g., BM25, TF-IDF) and single-step dense retrievers (e.g., Contriever, DPR) exhibit the lowest latency (e.g., $< 1.0s$). Thought-Retriever incurs a moderate overhead (e.g., 1.85s on Abstract-single) compared to these methods. This additional cost stems from the generation-based nature of our approach, where the model must query the thought memory and potentially generate new thoughts, rather than simply matching static vectors. However, given the significant performance gains demonstrated in the main results, this latency increase is acceptable for most applications.

**Comparison with Advanced Retrieval and Long-Context Baselines.** Critically, Thought-Retriever demonstrates superior efficiency against advanced reasoning-heavy and long-context baselines:

- **vs. IRCoT:** Iterative retrieval methods like IRCoT suffer from high latency due to their multi-step "retrieve-then-reason" loop. For instance, on the complex *Abstract-multi* dataset, IRCoT requires 10.50s per query. In contrast, Thought-Retriever achieves a $3\times$ speedup (3.20s) by leveraging pre-generated thoughts directly from memory, avoiding repetitive retrieval cycles at inference time.

- **vs. Long-Context LLMs:** Processing ultra-long contexts imposes a heavy computational burden on the attention mechanism. The long-context model *Nous Hermes-32k* exhibits the highest latency, reaching 28.50s on *Gov Report*. Thought-Retriever bypasses this bottleneck by retrieving concise, high-level thoughts instead of raw, long documents, reducing the input token load significantly and achieving an $8\times$ speedup (3.50s) on the same dataset.

**Conclusion.** Thought-Retriever strikes a favorable balance between efficiency and performance. It significantly outperforms fast but weak retrievers in reasoning capability, while remaining much faster than computationally expensive long-context models and iterative agents.

## S   Sequential Interaction Case Study: Visualizing Self-Evolution

To address the qualitative dynamics of how Thought-Retriever interacts with different user queries to achieve "self-evolution," we present a sequential case study using our Arxiv Copilot system.

It is important to clarify that in our framework, *self-evolution* refers to the continuous expansion and refinement of the Thought Memory ($\mathcal{T}$), as defined in Algorithm 1, rather than the updating of LLM parameters. This qualitative analysis complements the quantitative "scaling law" of thoughts demonstrated in Figure 5.

**Scenario:** A user is researching the topic of "Hallucination Mitigation in LLMs." The system starts with an empty Thought Memory.

**Turn 1: Initial Knowledge Acquisition**

- *User Query:* "What are the primary causes of hallucinations in Large Language Models?"

- *Retrieval:* The system retrieves raw data chunks from the external knowledge base (e.g., papers discussing data bias and source-reference divergence).

- *Answer Generation:* The model explains that hallucinations often stem from a conflict between the model's parametric memory and the provided context, or noise in the training data.

- *Self-Evolution (Memory Update):* Based on this interaction, the system generates and stores a new thought:

  > *Thought $T_1$:* Hallucinations in LLMs are primarily caused by the divergence between internal parametric knowledge and external context, often exacerbated by noisy training data.

**Turn 2: Leveraging Past Thoughts**

- *User Query:* "How does Retrieval-Augmented Generation (RAG) attempt to solve generation errors?"

- *Retrieval:* The system retrieves raw chunks related to RAG. Crucially, it also retrieves *Thought $T_1$* because "generation errors" are semantically linked to "Hallucinations" in $T_1$.

- *Answer Generation:* Leveraging $T_1$, the model connects RAG's mechanism directly to the root cause identified in Turn 1. It explains that RAG aligns the "external context" (mentioned in $T_1$) with the generation process.

- *Self-Evolution (Memory Update):* The system synthesizes a deeper insight:

  > *Thought $T_2$:* RAG mitigates hallucinations by grounding generation in the retrieved external context. However, to be effective, it must ensure the retrieved context is relevant to avoid exacerbating the source-reference divergence.

**Turn 3: Handling Abstract/Application Queries**

- *User Query:* "Propose a robust architecture for a trustworthy medical AI assistant."

- *Retrieval:* The query is abstract and does not explicitly mention "hallucination" or "RAG." However, the semantic vector for "trustworthy" matches with *Thought $T_2$* (mitigating errors).

- *Result:* The system retrieves *Thought $T_2$* and immediately proposes a RAG-based architecture with strict relevance filtering.

- *Analysis:* Without the self-evolution in Turns 1 and 2, the system might have just retrieved generic papers on medical AI. By retrieving $T_2$, it applies the high-level principle (RAG + Relevance Check) derived from previous interactions, demonstrating how the system has "evolved" to handle more complex, application-oriented tasks efficiently.

# T Data Construction and Filtering Process

To ensure the high quality and academic rigor of **AcademicEval**, we followed a systematic data construction pipeline. The process consists of three main stages: collection, preprocessing, and filtering.

**1. Data Collection.** We initiated our data pool by sourcing raw research papers from arXiv, specifically targeting the Computer Science domain (e.g., cs. CL, cs. LG, cs.AI). To align with the capabilities of modern LLMs, we focused on recent high-quality publications to serve as the external knowledge source.

**2. Preprocessing and Parsing.** Since raw PDFs contain noise (headers, footers, citation indices) that can disrupt LLM ingestion, we employed a parsing pipeline to convert PDF documents into a structured text format. This step allows us to cleanly separate the *main body* (used as external knowledge context) from the *abstract* (used as ground truth), while effectively removing non-textual elements.

**3. Filtering Criteria.** From the parsed corpus, we applied strict filtering criteria to select the final test set for AcademicEval:

- **Length Constraint:** To strictly evaluate long-context capabilities (a core motivation of Thought-Retriever), we filtered for papers that exceed a substantial token threshold. This ensures that the retrieval and reasoning challenge is non-trivial.

- **Structure Integrity:** We selected papers with clear section demarcations to ensure consistent evaluation formats.

**4. Test Set Sampling.** Crucially, consistent with our analysis in Section 4.4, we performed **stratified sampling** based on the abstract's abstraction level. This strategy ensures a balanced representation of difficulty levels across the benchmark, allowing us to evaluate the model's performance on both fact-based and reasoning-heavy queries.

**Original Abstract:**

Benchmarks such as GLUE have helped drive advances in NLP by incentivizing the creation of more accurate models. While this leaderboard paradigm has been remarkably successful, a historical focus on performance-based evaluation has been at the expense of other qualities that the NLP community values in models, such as compactness, fairness, and energy efficiency. In this opinion paper, we study the divergence between what is incentivized by leaderboards and what is useful in practice through the lens of microeconomic theory. We frame both the leaderboard and NLP practitioners as consumers and the benefit they get from a model as its utility to them. With this framing, we formalize how leaderboards – in their current form – can be poor proxies for the NLP community at large. For example, a highly inefficient model would provide less utility to practitioners but not to a leaderboard, since it is a cost that only the former must bear. To allow practitioners to better estimate a model's utility to them, we advocate for more transparency on leaderboards, such as the reporting of statistics that are of practical concern (e.g., model size, energy efficiency, and inference latency).

**Abstract by Thought Retriever:**

The paper "Utility is in the Eye of the User: A Critique of NLP Leaderboards" critiques the prevalent assumption that leaderboard metrics accurately reflect the utility of NLP models. It emphasizes the need to consider multiple factors beyond predictive accuracy, including energy efficiency, model size, and inference latency, when evaluating NLP models. The paper argues that leaderboard values often diverge from the values of practitioners, calling for increased transparency on leaderboards to better align these metrics with the practical concerns of the NLP community. It highlights the importance of a holistic approach to NLP evaluation, advocating for the inclusion of practical statistics on leaderboards to provide a more comprehensive measure of model utility. The research suggests that this shift towards transparency and comprehensive evaluation will encourage the development of models that are not only accurate but also efficient and fair, ultimately better serving the needs of diverse users in the NLP community.

. . . . . . . . . . . . . . . . . . . . . . . . . . . . . . . . . . . . . . . . . . . . . . . . . . . . . . . . . . . . . . . . . . . . . . . . . . . . . . .

**Comments from Expert LLMs:**

This abstract effectively **captures all critical aspects** of the original while emphasizing the divergence between leaderboard metrics and practical utility. It **maintains clarity and completeness**, making it an excellent summary of the paper's main arguments. Specifically, it includes key elements such as energy efficiency, model size, and the need for a holistic approach, which are well-aligned with the original abstract. The Thought Retriever abstract is comprehensive and aligns well with the original content, making it **the best among the generated abstracts.**

Figure 15: **Qualitative Example - Original Abstract and Abstract Generated by Thought-Retriever.** This figure presents example outputs from different methods using data from the AcademicEval-abstract-single dataset. Specifically, it shows the original abstract alongside the abstract generated by Thought-Retriever, accompanied by a comment from an expert LLM. Comparison examples generated by DPR and TF-IDF are shown in Figure 16, while comparison examples by the long context model can be found in Figure 17.

**Abstract by DPR:**

The text discusses the importance of including costs like model size, energy efficiency, training time, and inference latency in leaderboard rankings for NLP models, which currently prioritize accuracy above all else. This focus on accuracy can result in models that provide high utility to leaderboards but not to practitioners, as they may come with unaccounted costs. The text suggests that model creators could voluntarily submit these cost statistics, but without a requirement, there is no explicit incentive to do so. It proposes creating user-specific leaderboards with desired thresholds and reporting performance on the worst-off group to incentivize improvements. The use of revealed preference theory is suggested to estimate a model's utility by examining its real-world use. The text also advocates for making usage statistics public and allowing users to adjust metric weightings to create personalized leaderboards, aligning rankings with practical utility.

**Comments from Expert LLMs:**

This abstract is detailed and covers the essential points but tends to be **repetitive and slightly verbose.** It effectively communicates the need for including various costs in leaderboard rankings, but it **lacks the concise clarity** found in the Thought Retriever abstract. Additionally, the abstract is somewhat **incoherent** due to the repetitive structure and lack of clear flow between ideas. It does **not explicitly mention the holistic approach to evaluation or the practical applications of this research**, which are important elements from the original abstract.

**Abstract by TFIDF:**

Recent research has focused on the evaluation and development of NLP models, highlighting issues such as brittleness, bias, and lack of robustness. Datasets like SQuAD 2.0 have addressed challenges like unanswerable questions, while biases in NLP systems lead to better performance on gender-stereotypical inputs and inanimate objects. A key issue in evaluation is aligning priorities between leaderboard rankings and practitioners' preferences. While accuracy is valued by both, practitioners also consider out-of-distribution data, fairness, and real-world applicability, which leaderboards often ignore. The study suggests that leaderboard design should include prediction costs and fairness measures to better align with real-world requirements. Alternative evaluation methods that reflect the complexities of NLP model usage are also recommended.

**Comments from Expert LLMs:**

This abstract is concise and well-organized, addressing the key points effectively. It communicates the need for better alignment between leaderboard rankings and practical utility but **does not delve as deeply** into specific suggestions for improvement as the Thought Retriever abstract. While it provides a detailed discussion on issues like brittleness and bias, it might be **too detailed**, potentially overwhelming the reader with **too many aspects without a clear focus on the main argument of the original abstract.** Additionally, it **lacks specific elements** like the advocacy for increased transparency and the detailed explanation of how leaderboards can be poor proxies for the NLP community, which are crucial parts of the original abstract.

Figure 16: **Qualitative Example - Abstracts Generated by DPR and TF-IDF.** This figure presents example outputs using data from the AcademicEval-abstract-single dataset, generated by traditional methods: DPR and TF-IDF. We also include comments from an expert LLM. The original abstract and the abstract generated by our Thought-Retriever can be found in Figure 15.

**Abstract by Nours Hermes - 32k:**

The text addresses the limitations of performance-based evaluation in NLP tasks and advocates for more comprehensive leaderboard designs. It points out the disparity between leaderboard priorities and the values of the NLP community, such as fairness, energy efficiency, and robustness. By viewing leaderboards and NLP practitioners as consumers of models, the text suggests using utility functions to align their preferences. Three main limitations are identified: non-smooth utility, prediction cost, and robustness. Leaderboards only increase utility from accuracy improvements that affect rankings, while practitioners value consistent accuracy gains. Leaderboards ignore prediction costs, such as model size, energy efficiency, and inference latency. They also overlook robustness, crucial for real-world applications. To address these issues, the text proposes that leaderboards require transparency, reporting statistics like training time, model size, inference latency, and energy efficiency. This transparency would help practitioners estimate model utility and re-rank models based on their needs. The text also suggests personalized leaderboards, allowing users to adjust metric weightings or creating leaderboards tailored to different user types based on their preferences or actual usage.

- - - - - - - - - - - - - - - - - - - - - - - - - - - - - - - - - - - - - - - - - - - - - - - - - - - - - - - - - - -

**Comments from Expert LLMs:**

The long context model's abstract is detailed but **overly verbose**, making it **harder to quickly grasp the core arguments**. In contrast, the thought retriever's abstract is more succinct and focused, effectively summarizing the critique of leaderboard metrics and the need for comprehensive evaluation factors. This makes the thought retriever's abstract clearer and better aligned with the original abstract's intent.

Figure 17: **Qualitative Example - Abstracts Generated by Long Context Model.** This figure presents example outputs using data from the AcademicEval-abstract-single dataset, generated by the long context model Nours Hermes - 32k. The original abstract and the abstract generated by our Thought-Retriever can be found in Figure 15.

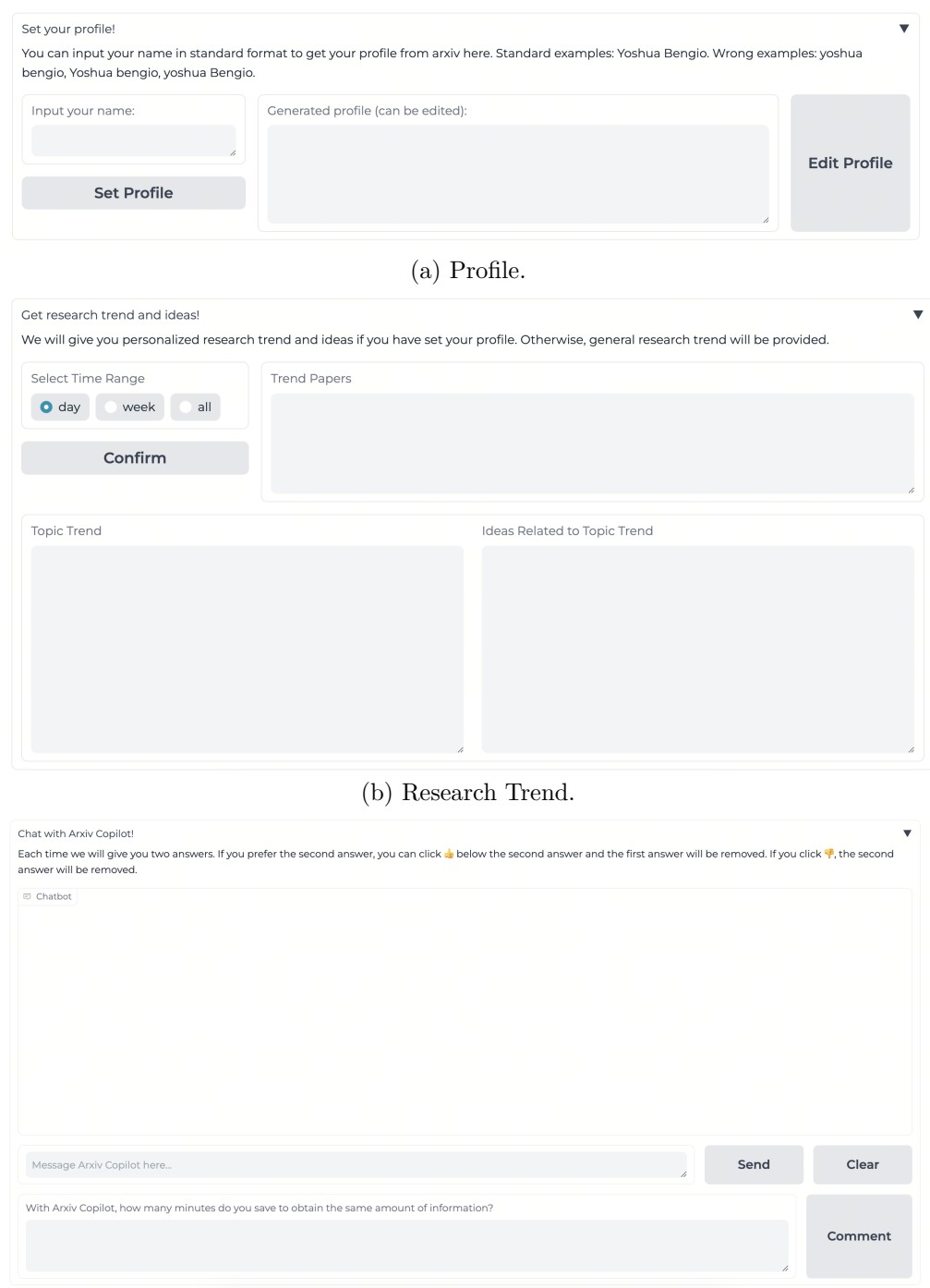

(a) Profile.

(b) Research Trend.

(c) Chat and Feedback.

Figure 18: **Arxiv Copilot Demo.** This figure shows the demo built based on our proposed Thought-Retriever, which is publicly available on Hugging Face. It offers personalized academic services, aiming to test the real-world robustness of our algorithm and provide social benefits.

