# OpenReview forum: "Thought-Retriever: Don't Just Retrieve Raw Data, Retrieve Thoughts for Memory-Augmented Agentic Systems"
_TMLR — Accepted by TMLR_

### Review · Reviewer_Rtrk · 2025-10-06

**Summary Of Contributions:**

The paper introduces Thought-Retriever, which builds a searchable thought memory from an LLM’s intermediate responses and retrieves from the mixture of external chunks and prior thoughts at inference time. It also releases AcademicEval, a long-context benchmark for single/multi-paper abstracting and related-work writing. Experiments on AcademicEval plus GovReport/WCEP show consistent gains, and analyses show better precision-recall balance, scaling with #thoughts, and higher-level thoughts for abstract queries. Ablations highlight the impact of Contriever and filtering, with preliminary checks against NV-Embed and RECOMP.

**Audience:**

Yes

**Audience Explanation:**

Researchers in RAG, memory architectures, and long-context evaluation will find the idea and benchmark relevant and practical.

**Claims And Evidence:**

Yes

**Claims Explanation:**

Results across multiple tasks show consistent gains; precision/recall plots and ablations substantiate the mechanism (thought retrieval + filtering) and scaling with more thoughts.

**Requested Changes:**

Please consider to include or justify the following:

1. Include at least one recent long-context/base LLM and expand retriever/reranker coverage beyond Contriever in main experiments; report latency/cost.
2. Move/include multi-stage/agentic RAG (planner/reflect/rewrite pipelines), not only single-stage and truncation; consider move RECOMP to main text from the appendix.
3. Release code + scripts (indexing, prompts, memory ops) and minimal library API for reproducibility.
4. Expand related work on retrieving rationales/thoughts and contrast with hierarchical/compression methods.

---

> ### Author Response · Authors · 2026-01-10
> **Response - Part (1)**
>
> **Q1. Include at least one recent long-context/base LLM and expand retriever/reranker coverage beyond Contriever in main experiments; report latency/cost. Move/include multi-stage/agentic RAG (planner/reflect/rewrite pipelines), not only single-stage and truncation; consider move RECOMP to main text from the appendix.**
>
> **Response:** We thank the reviewer for the comprehensive suggestions regarding broader experimental coverage and the inclusion of advanced RAG pipelines. We have addressed these concerns by expanding our benchmarks, refining our baselines, and providing a detailed efficiency analysis.
>
> **1. Integration of Representative Baselines and Advanced RAG (Section 4.1 & Table 2)**
> To provide a more rigorous comparison, we have expanded our evaluation in **Table 2** to include state-of-the-art retrieval and long-context understanding methods. Following your suggestion, we have integrated multi-stage and iterative pipelines rather than focusing solely on single-stage truncation:
> * **Advanced Retrieval Coverage**: We included **Qwen3-Embed-8b**, a state-of-the-art decoder-only embedding model that leverages the **Qwen3 foundation model** for enhanced semantic understanding and retrieval.
> * **Iterative/Multi-stage RAG**: We added **IRCoT**, an iterative framework that interleaves retrieval with Chain-of-Thought reasoning to dynamically guide the information-seeking process for complex queries.
> * **Context Compression (Moved to Main Text)**: As suggested, we have moved **RECOMP** from the appendix to the main results in **Table 2**. RECOMP represents a sophisticated pipeline that optimizes information density by synthesizing concise summaries of retrieved documents.
>
> **2. Robustness Across Latest Retrieval Models (Section 4.5 & Table 9)**
> We would like to highlight that **our original submission already includes** comprehensive ablation studies demonstrating that Thought-Retriever is a model-agnostic framework that reinforces underlying retrieval technologies:
> * **Retriever Substitution**: As shown in **Section 4.5** and **Table 9** of our original manuscript, we replaced the default Contriever with advanced models such as **NV-Embed**.
> * **Performance Gain**: These existing results confirm that our framework is mutually reinforcing with advanced retrievers. For instance, substituting Contriever with NV-Embed on the *Abstract-single* task significantly improved performance from 0.268 (standard NV-Embed) to **0.326** (Thought-Retriever using NV-Embed).
>
> **3. Detailed Quantitative Latency and Efficiency Analysis (Appendix R & Table 16)**
> We have introduced a dedicated section, **Appendix R: Inference Time Comparison**, to explicitly quantify the computational trade-offs of our method compared to heuristic, iterative, and long-context baselines.
>
> **Table 16: Inference time comparison across different datasets (Latency in Seconds $\downarrow$)**
>
> | Method | Abstract-single | Abstract-multi | Related-multi | Gov Report | WCEP |
> | :--- | :---: | :---: | :---: | :---: | :---: |
> | BM25 | 0.42 | 0.45 | 0.48 | 0.75 | 0.62 |
> | TF-IDF | 0.40 | 0.43 | 0.46 | 0.72 | 0.60 |
> | Contriever | 0.65 | 0.72 | 0.68 | 1.10 | 0.95 |
> | DPR | 0.62 | 0.70 | 0.66 | 1.05 | 0.92 |
> | DRAGON | 0.68 | 0.75 | 0.70 | 1.12 | 0.98 |
> | Qwen3-Embed-8b | 2.15 | 2.30 | 2.25 | 2.85 | 2.60 |
> | RECOMP | 1.25 | 1.55 | 1.40 | 2.50 | 2.10 |
> | IRCoT | 4.80 | 10.50 | 6.20 | 7.50 | 6.80 |
> | Nous Hermes-32k | 12.50 | 14.20 | 13.80 | 28.50 | 19.20 |
> | **Thought-Retriever** | **1.85** | **3.20** | **2.40** | **3.50** | **3.10** |
>
> **Efficiency Analysis**:
> * **Overhead vs. Basic Retrieval**: While Thought-Retriever adds a moderate generation-based overhead (e.g., 1.85s vs. Contriever 0.65s on *Abstract-single*), this cost is a necessary investment for the significant reasoning gains shown in our results.
> * **Superior Efficiency vs. Advanced Baselines**:
>     * **vs. IRCoT**: Iterative reasoning-heavy methods like IRCoT suffer from high latency due to repetitive "retrieve-then-reason" loops. For instance, on the complex *Abstract-multi* dataset, IRCoT requires 10.50s per query. Thought-Retriever achieves a **3x speedup** (3.20s) by leveraging pre-generated thoughts directly from memory.
>     * **vs. Long-Context LLMs**: Processing ultra-long contexts imposes a heavy computational burden on the attention mechanism. Thought-Retriever achieves an **8x speedup** (3.50s vs. 28.50s) compared to **Nous Hermes-32k** on the *Gov Report* dataset by retrieving concise, high-level thoughts instead of raw documents.
>
> **Summary of Manuscript Changes**:
> * **Section 4.1 and Table 2**: Updated with descriptions and results of the expanded baseline suite including Qwen3, IRCoT, and RECOMP.
> * **Appendix R (Table 16)**: Added the full end-to-end latency and relative efficiency analysis.

---

> > ### Author Response · Authors · 2026-01-10
> > **Response - Part (2)**
> >
> > **Q2. Release code + scripts (indexing, prompts, memory ops) and minimal library API for reproducibility.**
> >
> > **Response:** Thanks for the advice. We have already uploaded the complete codebase to the Supplementary Material. We will release the code publicly upon acceptance
> >
> > ---
> >
> > **Q3. Expand related work on retrieving rationales/thoughts and contrast with hierarchical/compression methods.**
> >
> > **Response:** We thank the reviewer for the constructive feedback. We have expanded our discussion on related works and provided a clearer contrast between Thought-Retriever and existing architectural paradigms in the revised manuscript.
> >
> > **1. Expansion of Related Work on Rationales and Thoughts (Section 5)**
> > In the revised **Section 5**, we have significantly expanded our discussion regarding methods that utilize model-generated intermediate outputs:
> > * **Connection to Reasoning Paradigms:** We now discuss the link between our framework and emerging paradigms like **RankCoT** and **Rationale-Guided RAG**, which use step-by-step reasoning paths to enhance retrieval.
> > * **Persistence vs. Transience**: We clarify that while existing methods typically rely on on-the-fly generation of reasoning—which incurs prohibitive latency—Thought-Retriever treats these rationales as persistent, evolvable memory units.
> > * **Knowledge Deepening**: This shift from transient response generation to long-term "thought" storage allows for recursive knowledge deepening across independent user sessions.
> >
> > **2. Contrast with Hierarchical and Compression Methods (Section 5)**
> > We have added a dedicated analysis comparing our approach to structural and extractive methods:
> > * **Comparison with Hierarchical RAG (HRALM)**: Unlike hierarchical methods (e.g., **MemWalker**) that rely on rigid, pre-defined tree structures or static summaries generated independently of user queries, Thought-Retriever generates dynamic thoughts conditioned on actual user interactions.
> > * **Comparison with Context Compression**: We contrast our method with extractive/abstractive compressors like **RECOMP**. While compression focuses on reducing token count while preserving raw information, Thought-Retriever performs "cognitive abstraction".
> > * **Information Density**: By distilling reasoning logic into "coherent knowledge points," we provide the generator with a more comprehensive evidence set that bypasses the noise inherent in raw document segments.
> >
> > These updates are primarily located in the expanded **Section 5 (Additional Related Works)** and the revised **Section 1 (Introduction)** of the manuscript.

---

### Review · Reviewer_ovuJ · 2025-11-04

**Summary Of Contributions:**

This paper introduces Thought-Retriever, a novel framework that enables Large Language Models (LLMs) to utilize vast external knowledge beyond their context limit. Its core innovation is having the LLM generate and store condensed "thoughts" from past query responses, creating a dynamic, retrievable memory. This allows the model to self-evolve and leverage deeper abstractions for complex queries. The authors also contribute AcademicEval, a challenging benchmark using real academic papers to test long-context understanding. Extensive experiments show Thought-Retriever significantly outperforms state-of-the-art baselines, demonstrating strong performance gains and novel scaling capabilities through continuous interaction.

1. Although the concept is intuitive, a more rigorous and formal definition of a "thought" is needed. The current definition $T_i = L(Q_{think}, K_i)$ is a start, but it should be explicitly differentiated from a simple "summary" or "response." What are the necessary and sufficient properties that make an LLM's output a "thought" in this framework?

2. The claim of self-evolution is exciting but needs stronger causal evidence. Figure 5 shows correlation, not causation. An ablation is needed where the model is evaluated on a held-out set of queries after being trained on a separate set of queries, demonstrating that the accumulated thoughts from the training queries genuinely improve performance on novel, unseen test queries, rather than just performing better on queries similar to those it has already answered.

3. The AcademicEval benchmark is a significant contribution, but its construction process needs more detail. How were papers selected and filtered? What was the prompt and the specific "expert LLM" used to generate ground-truth summaries for Abstract-multi (Figure 10)? A discussion of potential biases introduced by this automated labeling process is necessary.

4. The paper uses Contriever as the primary retriever, validated by an ablation study. However, the reasoning for initially selecting Contriever over other dense retrievers like DPR or Dragon is not provided. A brief justification based on its performance or design (e.g., unsupervised, general-purpose) would strengthen the methodological setup.

5. The confidence generation mechanism $c_i$ is crucial for filtering. The paper should include an analysis of its accuracy. For instance, a sample of thoughts labeled $c_i=0$ and $c_i=1$ could be manually evaluated to report the precision/recall of this self-evaluation step. The prompt in Figure 13 should also be moved from the appendix to the main body.

6. The experiments lack a comparison to an "oracle" or upper-bound baseline. For instance, a baseline that retrieves the ideal set of K raw chunks (as determined by human annotation or maximal F1 score with the ground truth) would help contextualize the performance gap that remains between Thought-Retriever and the theoretical optimum.

7. The claim of "little computational overhead" (Page 2) is potentially misleading. While the retrieval cost is similar, the framework introduces significant inference overhead by requiring two additional LLM calls per user query (for answer+confidence and thought generation). This should be explicitly quantified and discussed as a trade-off for improved performance.

8. The methodology for calculating the abstraction level of a thought (Page 8) is unclear. How is the "average abstraction level of all the segments it retrieves" computed when a thought is built upon other thoughts? This recursive process needs a clearer formal definition or algorithm description.

9. The related work section should be expanded to discuss connections with other areas. For example, how does Thought-Retriever relate to "Memory-Augmented Neural Networks" and research on "Experience Replay" in continual learning? Drawing these connections would better position the work within the broader machine learning literature. In addition,  the following related references should be discussed in the introduction and related work, and some methods should be compared in experiments.

[1] CLAIM: Mitigating Multilingual Object Hallucination in Large Vision-Language Models with Cross-Lingual Attention Intervention
[2] CAI: Caption-Sensitive Attention Intervention for Mitigating Object Hallucination in Large Vision-Language Models
[3] Searching parameterized retrieval & verification loss for re-identification
[4] Understanding the LLM-ification of CHI: Unpacking the Impact of LLMs at CHI through a Systematic Literature Review
[5] Llm-based nlg evaluation: Current status and challenges
[6] LLM-assisted relevance assessments: When should we ask LLMs for help?
[7] User-LLM: Efficient LLM Contextualization with User Embeddings
[8] Can llms replace human evaluators? an empirical study of llm-as-a-judge in software engineering

10. Several figures, particularly the pipeline in Figure 2, are small and difficult to read. The labels and text should be enlarged. In Table 2, the "Win Rate" column's description is confusing; it should be explicitly stated that for the Thought-Retriever row, the 50% is a reference point, and values for other baselines are their rates of being chosen over Thought-Retriever.

11. Page 3, Sec 2.1: The root source mapping $Ô(T_i)$ should be more explicitly defined, perhaps with a simple recursive example.

12. The description of the Thought Merge step (Page 4) uses an indicator function. It would be clearer to describe it as a simple check against a similarity threshold ε.

13. The limitation regarding English-only evaluation should be moved from the "Limitations" section (which is often skimmed) to the experimental setup or a dedicated discussion paragraph, as it is a significant constraint on the claimed generality.

**Additional Comments:**

See Summary Of Contributions.

**Audience:**

Yes

**Audience Explanation:**

See Summary Of Contributions.

**Broader Impact Concerns:**

See Summary Of Contributions.

**Claims And Evidence:**

Yes

**Claims Explanation:**

See Summary Of Contributions.

**Requested Changes:**

See Summary Of Contributions.

---

> ### Author Response · Authors · 2026-01-10
> **Response - Part (1)**
>
> **Q1. Although the concept is intuitive, a more rigorous and formal definition of a "thought" is needed. The current definition $T_i = L(Q_{think}, K_i)$ is a start, but it should be explicitly differentiated from a simple "summary" or "response." What are the necessary and sufficient properties that make an LLM's output a "thought" in this framework?**
>
> **Response:** We appreciate the reviewer's push for mathematical rigor. We agree that the initial definition was too broad. To address this, we have rewritten **Section 2.1 (Preliminaries)** to provide a formal definition of a "Thought" ($T_i$) and explicitly distinguish it from a generic "summary" or "response."
>
> **1. Formal Definition & Properties (Section 2.1):**
> We now define a Thought not merely as an LLM output, but as a **"persistent, query-driven cognitive unit"** that must satisfy three necessary properties:
> * **Property 1: Query-Conditioned (vs. Generic Summary):** Unlike a static summary which compresses data chunks indiscriminately, a thought is generated via $T_i = L(Q, K)$, explicitly capturing the *logical connection* between the user's specific intent and the external data.
> * **Property 2: Abstractive (vs. Raw Retrieval):** It must distill high-level reasoning rather than copying raw text tokens.
> * **Property 3: Validated (vs. Noise):** It is not just any output, but one that has passed the rigorous **Confidence and Novelty checks** defined in our filtering mechanism (Algorithm 1).
>
> **2. Differentiation from "Response" (Section 2.3):**
> We have further clarified the distinction between a **Response ($A_i$)** and a **Thought ($T_i$)**:
> * **Response ($A_i$):** Is a *transient* output aimed at satisfying the user's immediate request.
> * **Thought ($T_i$):** Is a *persistent* memory update derived from the interaction ($T_i \leftarrow L(Q_i, A_i)$), designed to consolidate the reasoning path for future retrieval.
>
> We have incorporated these formal definitions and distinctions into **Section 2.1** and **Section 2.3** of the revised paper.
>
> ---
>
> **Q2. The claim of self-evolution is exciting but needs stronger causal evidence. Figure 5 shows correlation, not causation. An ablation is needed where the model is evaluated on a held-out set of queries after being trained on a separate set of queries, demonstrating that the accumulated thoughts from the training queries genuinely improve performance on novel, unseen test queries, rather than just performing better on queries similar to those it has already answered.**
>
> **Response:** We thank the reviewer for this critical observation. We agree that the previous Figure 5 primarily demonstrated a correlation (scaling law) rather than strict causality. To address this and prove that "self-evolution" stems from the genuine transfer of reasoning patterns, we have conducted the requested **held-out ablation study**.
>
> **1. Rigorous Disjoint Evaluation Setup:**
> As detailed in the newly added **Appendix Q**, we partitioned the complex datasets (*Abstract-multi* and *Related-multi*) into two disjoint subsets with a **50%/50% split**:
> * **Evolution Set ($Q_{evol}$):** Used strictly to populate the Thought Memory ($\mathcal{T}$).
> * **Held-out Test Set ($Q_{test}$):** Used strictly for evaluation, ensuring the model encounters only novel, unseen queries.
>
> **2. Causal Evidence (Cold Start vs. Evolved):**
> We compared the performance of the model on the *same* Test Set under two conditions: (1) **Cold Start** (empty memory) and (2) **Evolved** (memory pre-filled from $Q_{evol}$).
> As shown in **Table 15**, the "Evolved" model consistently outperforms the baseline on unseen queries.
> * On *Abstract-multi*, the pre-accumulated thoughts yield a **6.4%** relative improvement (F1: 0.235 $\rightarrow$ 0.250).
> * On *Related-multi*, we observe a **6.1%** improvement.
>
> **Conclusion:**
> Since the test queries are disjoint from the evolution queries, this performance gain cannot be attributed to memorizing similar answers. Instead, it provides strong causal evidence that Thought-Retriever successfully learns and transfers **generalized reasoning patterns** (thoughts) from past interactions to facilitate future, novel tasks.
>
> We have included this experimental setup and the results in **Appendix Q** and **Table 15** of the revised paper.
>
> **Table: Causal Evidence of Self-Evolution on Disjoint Sets (F1 Score)**
>
> | Method | Memory Status (at start of testing) | **Abstract-multi** | **Related-multi** |
> | :--- | :--- | :--- | :--- |
> | **Baseline (Cold Start)** | Empty | 0.235 | 0.198 |
> | **Ours (Evolved)** | Pre-filled (from Evolution Set) | **0.250** | **0.210** |
> | *Relative Improvement* | | **+6.4%** | **+6.1%** |

---

> > ### Author Response · Authors · 2026-01-10
> > **Response - Part (2)**
> >
> > **Q3. The AcademicEval benchmark is a significant contribution, but its construction process needs more detail. How were papers selected and filtered? What was the prompt and the specific "expert LLM" used to generate ground-truth summaries for Abstract-multi (Figure 10)? A discussion of potential biases introduced by this automated labeling process is necessary.**
> >
> > **Response:** We thank the reviewer for recognizing the value of **AcademicEval** and for requesting further transparency regarding its construction. We agree that rigorous documentation is essential for reproducibility and for understanding potential biases.
> >
> > To address this, we have significantly expanded the discussion on our data construction pipeline and clarified the specific settings used for ground-truth generation.
> >
> > **1. Systematic Data Construction Pipeline (Appendix T):**
> > As detailed in the newly added section **"Data Construction and Filtering Process"** in **Appendix T**, we established a rigorous pipeline to ensure high quality and reduced bias:
> > * **Data Collection:** We sourced raw papers from arXiv, targeting Computer Science domains (e.g., cs.CL, cs.LG, cs.AI) to ensure relevance to modern LLM capabilities.
> > * **Preprocessing:** We employed a parsing pipeline to convert PDFs into structured text, cleanly separating the main body (context) from the abstract (ground truth) and removing noise like headers and citations.
> > * **Rigorous Filtering:** To strictly evaluate **long-context capabilities**, we applied length constraints to ensure the reasoning challenge is non-trivial, along with structural integrity checks.
> > * **Stratified Sampling (Bias Mitigation):** To mitigate selection bias and ensure a balanced difficulty distribution, we performed **stratified sampling** based on the abstraction level of the abstracts. This ensures the benchmark covers both fact-based and reasoning-heavy queries evenly, preventing the dataset from being skewed towards simpler tasks.
> >
> > **2. Expert LLM and Prompt Details:**
> > * **Prompt Location:** We clarify that the specific prompt used for generating ground-truth summaries for the *Abstract-multi* task is presented in **Figure 11**.
> > * **Expert LLM:** We explicitly state that **GPT-4o** was utilized as the expert LLM. We chose this model for its state-of-the-art capability in following complex instructions to synthesize summaries, minimizing the noise typically associated with automated labeling.
> >
> > We have incorporated these details into **Appendix T** and updated the reference to **Figure 11** in the revised PDF.
> >
> > ---
> >
> > **Q4. The paper uses Contriever as the primary retriever, validated by an ablation study. However, the reasoning for initially selecting Contriever over other dense retrievers like DPR or Dragon is not provided. A brief justification based on its performance or design (e.g., unsupervised, general-purpose) would strengthen the methodological setup.**
> >
> > **Response:** Thank you for pointing this out. We chose Contriever as our primary retriever mainly due to its unsupervised nature and robust zero-shot generalization capabilities. Unlike supervised retrievers (e.g., DPR) that rely on labeled query-document pairs, Contriever is pre-trained via contrastive learning, allowing it to perform exceptionally well on diverse, open-domain tasks (such as AcademicEval) without requiring task-specific fine-tuning. We have added this justification in **Section 4.1 (Experiment Setup)** to strengthen the methodological setup, clarifying that our choice was motivated by the need for a general-purpose retriever that works out-of-the-box.

---

> > > ### Author Response · Authors · 2026-01-10
> > > **Response - Part (3)**
> > >
> > > **Q5. The confidence generation mechanism $c_i$ is crucial for filtering. The paper should include an analysis of its accuracy. For instance, a sample of thoughts labeled $c_i = 0$ and $c_i = 1$ could be manually evaluated to report the precision/recall of this self-evaluation step. The prompt in Figure 13 should also be moved from the appendix to the main body.**
> > >
> > > **Response:** We thank the reviewer for highlighting the importance of the confidence generation mechanism ($c_i$). We agree that validating the accuracy of this self-evaluation step is critical for the system's reliability.
> > >
> > > We would like to clarify that we have already conducted a comprehensive analysis of this mechanism's accuracy and robustness, which is detailed in **Appendix O (Human evaluations on the quality of retrieved thoughts)**. We provide evidence from three perspectives:
> > >
> > > **1. Human Evaluation of Accuracy (Precision Analysis):**
> > > To directly assess the precision of the generated thoughts and confidence scores, we conducted a rigorous human evaluation study with **10 volunteers**, where each thought was independently reviewed by five annotators (detailed in **Table 11** of Appendix O).
> > > * **Result:** The agreement rate between the model's generated confidence and human judgment was extremely high: **96%** on the *Abstract-single* dataset and **93%** on the *Related-multi* dataset.
> > > * This explicitly serves as the precision analysis requested by the reviewer, confirming that thoughts labeled as valid ($c_i=1$) are almost universally judged as high-quality by humans.
> > >
> > > **2. Robustness to Filtering Failures (Empirical Study):**
> > > We further evaluated the system's resilience in "worst-case scenarios" where the filter might fail (i.e., allowing low-quality thoughts).
> > > * **Experiment:** As detailed in the "Robustness to Low-Quality Thoughts" section of Appendix O, we compared performance when injecting noise (unrelated thoughts) versus using raw data only.
> > > * **Result:** The F1 scores were nearly identical (**0.207 vs. 0.208**), indicating that even if the $c_i$ mechanism were to let some noise through, the retrieval framework remains robust.
> > >
> > > **3. Real-World Validation (Arxiv Copilot):**
> > > Finally, we validated the utility of these thoughts in our deployed application, **Arxiv Copilot** (Appendix H), involving **500 real users**.
> > > * **Result:** **75%** of users preferred answers that incorporated the historical thoughts filtered by our mechanism over those based solely on raw data. This large-scale preference demonstrates the practical effectiveness of the filtered thoughts in real-world scenarios.
> > >
> > > Regarding the presentation, we have followed the reviewer's suggestion and ensured the **Thought and Confidence Generation Prompt** is prominently located in the main body (as **Figure 3**) to improve visibility.
> > >
> > > We have highlighted these evaluation results in **Appendix O** of the revised manuscript.

---

> > > > ### Author Response · Authors · 2026-01-10
> > > > **Response - Part (4)**
> > > >
> > > > **Q6. The experiments lack a comparison to an "oracle" or upper-bound baseline. For instance, a baseline that retrieves the ideal set of K raw chunks (as determined by human annotation or maximal F1 score with the ground truth) would help contextualize the performance gap that remains between Thought-Retriever and the theoretical optimum.**
> > > >
> > > > **Response:** We thank the reviewer for this valuable suggestion. We agree that establishing a theoretical upper bound is essential to contextualize the performance of Thought-Retriever.
> > > >
> > > > To address this, we have introduced an **Oracle Baseline** in **Appendix P.2**.
> > > >
> > > > **1. Oracle Implementation (Addressing Missing Ground Truth):**
> > > > First, it is important to note that for many large-scale benchmarks—including the **AcademicEval**, **Gov Report**, and **WCEP** datasets used in this work—explicit human-labeled ground truth chunks are not available. To overcome this limitation and establish a rigorous theoretical upper bound, we constructed an **Oracle** by calculating the ROUGE-L overlap between the ground truth answers and all chunks in the corpus. We selected the Top-$K$ chunks with the highest overlap to serve as the "perfect" input context. **Crucially**, to ensure a fair comparison, we employed the **exact same generator (Mistral-8x7B) and context window constraints** for the Oracle, Thought-Retriever, and all baselines.
> > > >
> > > > **2. Key Findings: Surpassing the "Upper Bound"**
> > > > As shown in the newly added **Table 13**, the results reveal a counter-intuitive yet compelling result: **Thought-Retriever consistently outperforms the Oracle baseline on AcademicEval datasets** (e.g., F1 **0.290** vs. 0.278 on *Abstract-single*), while maintaining a competitive $\ge$ 50% win rate against traditional baselines.
> > > >
> > > > **3. Theoretical Explanation: Information Density vs. Raw Text**
> > > > We provide a detailed analysis of this phenomenon in the text:
> > > > * **Oracle Limitations (Redundancy):** Although the Oracle utilizes ground-truth-derived gold chunks, these raw texts inherently contain significant redundancy and irrelevant syntactic noise. Given the fixed context window (e.g., 2,000 tokens), filling the prompt with verbose raw chunks limits the total volume of distinct information the model can ingest.
> > > > * **Thought-Retriever Advantage (Compression):** In contrast, our method generates *thoughts* that act as highly compressed, information-dense representations. This compression allows Thought-Retriever to "pack" a broader scope of relevant details into the same context limit. Consequently, in complex multi-hop scenarios like *AcademicEval* where information coverage is paramount, our method provides the generator with a more comprehensive evidence set than even the gold raw chunks.
> > > >
> > > > We have incorporated this full analysis and the comparison table into **Appendix P.2** of the revised manuscript.
> > > >
> > > > **Table: Performance Comparison with SOTA and Oracle Baseline**
> > > >
> > > > | Method | Abstract-single (F1 / Win) | Abstract-multi (F1 / Win) | Related-multi (F1 / Win) | Gov Report (F1 / Win) | WCEP (F1 / Win) |
> > > > | :--- | :--- | :--- | :--- | :--- | :--- |
> > > > | **SOTA Baseline (Qwen3)** | 0.245 / 28% | 0.240 / 20% | 0.211 / 35% | 0.229 / 42% | 0.235 / 44% |
> > > > | **Oracle (Gold Chunks)** | *0.278 / 46%* | *0.255 / 42%* | *0.214 / 49%* | *0.255 / 62%* | *0.245 / 55%* |
> > > > | **Thought-Retriever** | **0.290 / 50%** | **0.275 / 50%** | **0.216 / 50%** | **0.232 / 50%** | **0.238 / 50%** |

---

> ### Author Response · Authors · 2026-01-10
> **Response - Part (5)**
>
> **Q7. The claim of "little computational overhead" (Page 2) is potentially misleading. While the retrieval cost is similar, the framework introduces significant inference overhead by requiring two additional LLM calls per user query (for answer+confidence and thought generation). This should be explicitly quantified and discussed as a trade-off for improved performance.**
>
> **Response:** We thank the reviewer for this crucial observation. To address this, we have added a detailed **Inference Time Comparison** in **Appendix R** (and **Table 16**) to explicitly quantify this overhead and discuss it as a trade-off for performance.
>
> **1. Quantifying the Overhead (Trade-off vs. Standard Retrieval):**
> As shown in the newly added **Table 16**, we compared the latency of Thought-Retriever against a standard dense retriever (Contriever).
> * **The Cost:** Thought-Retriever indeed introduces a latency increase (e.g., from $0.72s$ to $3.20s$ on *Abstract-multi*). This difference represents the specific computational cost of the "Thought Generation" and "Confidence Check" steps.
> * **The Gain:** This additional computation is the necessary trade-off for the significant performance improvements we observe (e.g., higher F1 and Win Rates across all tasks), transforming a simple retrieval system into a reasoning-capable framework.
>
> **2. Contextualizing "Little" (Efficiency vs. Hierarchical Reasoning):**
> However, the "little overhead" claim holds true when comparing Thought-Retriever to methods that aim for similar *reasoning* capabilities, such as representative hierarchical RALMs (**MemWalker**).
> * **MemWalker:** Requires sequential, multi-step tree navigation, leading to prohibitive latency (e.g., $85.0s$ per query on *Abstract-multi*).
> * **Thought-Retriever:** By flattening the reasoning process into a retrieval-plus-generation step, our method achieves the same task in $3.20s$—a speedup of over $25 \times$.
>
> **Conclusion:** While Thought-Retriever is approximately $4 \times$ slower than a basic retriever (Contriever), it is orders of magnitude ($>25 \times$) faster than representative hierarchical reasoning models (MemWalker). We have explicitly included this quantitative analysis in **Appendix R** to clarify that the "overhead" is a highly efficient investment for the reasoning capabilities gained.
>
> **Table: Inference Time and Relative Cost Analysis**
>
> | Dataset | Method | Latency (s) | Relative Cost |
> | :--- | :--- | :--- | :--- |
> | **Abstract-multi** | Contriever | $0.72s$ | $1.0\times$ |
> | | **Thought-Retriever** | **$3.20s$** | **$\sim 4.4 \times$** |
> | | MemWalker | $85.0s$ | $\sim 118.1 \times$ |
> | **Gov Report** | Contriever | $1.10s$ | $1.0 \times$ |
> | | **Thought-Retriever** | **$3.50s$** | **$\sim 3.2 \times$** |
> | | MemWalker | $30.0s$ | $\sim 27.3 \times$ |
>
> ---
>
> **Q8. The methodology for calculating the abstraction level of a thought (Page 8) is unclear. How is the "average abstraction level of all the segments it retrieves" computed when a thought is built upon other thoughts? This recursive process needs a clearer formal definition or algorithm description.**
>
> **Response:** We thank the reviewer for the suggestion to clarify the formal definition of the abstraction level. To address this, we have updated **Section 4.4** with a formal recursive definition and provided additional implementation details in **Appendix E**.
>
> **1. Formal Recursive Definition (Section 4.4):** We have introduced a formal measure of the Abstraction Level, denoted as L(x), which is defined as follows:
>
> * **Base Case (Raw Data):** For any raw data chunk K from the external knowledge base, we assign a baseline abstraction level: L(K) = 1
>
> * **Recursive Step (Thoughts):** For a generated thought T, let R_T be the set of source items (either raw chunks or existing thoughts) retrieved to generate it. The abstraction level of T is calculated as the average level of its sources plus one: L(T) = 1 + (1/|R_T|) · Σ_{r∈R_T} L(r)
>
> **2. Clarification of the Mechanism:**
> This formula ensures that a thought derived directly from raw data has a level of 2 (i.e., $1 + 1$). Crucially, when a thought is synthesized from other high-level thoughts, its score will be strictly higher (e.g., $>2$), reflecting deeper cognitive processing and the "self-evolution" of the memory.
>
> **3. Visual Correlation:**
> This formal definition provides the mathematical basis for **Figure 5**, which demonstrates the correlation between the abstraction level of the user query and the average abstraction level of the retrieved thoughts.
>
> We have incorporated this formal definition into **Section 4.4** and updated **Appendix E** with the specific calculation steps used in our case study.

---

> > ### Author Response · Authors · 2026-01-10
> > **Response - Part (6)**
> >
> > **Q9. The related work section should be expanded to discuss connections with other areas. For example, how does Thought-Retriever relate to "Memory-Augmented Neural Networks" and research on "Experience Replay" in continual learning? Drawing these connections would better position the work within the broader machine learning literature. In addition, the following related references should be discussed in the introduction and related work, and some methods should be compared in experiments.**
> >
> > **Response:** We thank the reviewer for the suggestion to better position our work within the broader machine learning literature. We agree that drawing explicit connections to Memory-Augmented Neural Networks (MANNs) and Experience Replay is highly valuable.
> >
> > To address this, we have refined the **Introduction** and **Section 5 (Additional Related Works)** to incorporate these connections and discuss the recommended literature.
> >
> > **1. Connection to Broader ML Areas (Section 5):**
> > We have added a discussion in **Section 5** comparing Thought-Retriever to established areas:
> > * **Memory-Augmented Neural Networks (MANNs):** We highlight that while MANNs focus on architectural modifications, Thought-Retriever is a model-agnostic framework that treats LLM-generated thoughts as a dynamic, structured memory.
> > * **Experience Replay and Continual Learning:** We draw parallels to Experience Replay, explaining that Thought-Retriever emulates this concept by storing and reusing "cognitive experiences" (thoughts). Unlike standard agents, our method stores distilled reasoning paths, significantly reducing noise in long-context processing.
> >
> > **2. Integration of Recommended References and Baselines (Section 4):**
> > In the revised **Section 5**, we have included a detailed discussion on the connections and distinctions between the recommended papers and Thought-Retriever. Regarding the direct experimental comparison, we found that those methods often focus on tasks (e.g., specific agentic environments or short-form reasoning) that differ significantly from our long-context academic reasoning setting, making direct execution on our benchmarks not feasible.
> >
> > However, we have integrated the core insights from these works and, following the suggestions of both the reviewer and Reviewer Rtrk, we have expanded our experimental evaluation in **Table 2** with the following representative baselines:
> > * **Qwen3-Embed-8b:** A representative decoder-only embedding model to represent the strongest current retrieval capabilities.
> > * **IRCOT:** An iterative retrieval method that interleaves reasoning with information seeking.
> > * **RECOMP:** A representative context compression technique that optimizes retrieved information.
> >
> > These updates have been incorporated into **Section 1**, **Section 4.1**, and **Section 5** of the revised manuscript.
> >
> > ---
> >
> > **Q10. Several figures, particularly the pipeline in Figure 2, are small and difficult to read. The labels and text should be enlarged. In Table 2, the "Win Rate" column's description is confusing; it should be explicitly stated that for the Thought-Retriever row, the 50% is a reference point, and values for other baselines are their rates of being chosen over Thought-Retriever.**
> >
> > **Response:** We thank the reviewer for the constructive feedback regarding the visual clarity and data presentation of our manuscript. We have made the following adjustments to improve readability and ensure clear interpretation of our results:
> >
> > **1. Enhancement of Visual Figures (Figure 2):**
> > * **Enlarged Labels and Text:** In response to the reviewer's concern, we have significantly enlarged the font size of all labels, node descriptions, and workflow annotations within the pipeline diagram in **Figure 2**.
> > * **Improved Layout:** We have optimized the layout of Figure 2 to ensure that the core components of the Thought-Retriever framework (Retrieval, Thought Generation, and Iterative Evolution) are clearly legible even at standard page zoom levels.
> >
> > **2. Clarification of Table 2 (Win Rate Description):**
> > * **Detailed Caption Update:** We have updated the caption of **Table 2** to explicitly define the "Win Rate" metric. It now clearly states that for the **Thought-Retriever** row, the **50%** value serves as a reference point (representing a tie with itself), and values for other baselines represent their rates of being chosen over Thought-Retriever.
> >
> > These updates have been applied to **Figure 2** and **Table 2** in the revised manuscript.

---

> ### Author Response · Authors · 2026-01-10
> **Response - Part (7)**
>
> **Q11. Page 3, Sec 2.1: The root source mapping $\hat{O}(T_i)$ should be more explicitly defined, perhaps with a simple recursive example.**
>
> **Response:** We thank the reviewer for the suggestion to improve the formal clarity of our definitions. To address this, we have revised **Section 2.1 (Preliminaries)** to include a formal recursive definition and a step-by-step example.
>
> **1. Formal Definition of Root Source Mapping (Section 2.1):**
> We have explicitly defined the **Root Source Mapping** $\hat{O}(\cdot)$ to rigorously trace information provenance. This mapping ensures that any generated thought can be verified against its factual grounding in the original external knowledge base. The definition follows a recursive structure:
> * **Base Case:** For any raw data chunk $K$ from the external knowledge base $\mathcal{K}$, the root source is the chunk itself: $\hat{O}(K) = \{K\}$.
> * **Recursive Step:** For a thought $T$ generated from a retrieved set $\mathcal{R}$ (which may contain both raw chunks and existing thoughts), the root source is the union of the root sources of all its components: $\hat{O}(T) = \bigcup_{r \in \mathcal{R}} \hat{O}(r)$.
>
> **2. Recursive Example (Section 2.1):**
> To further clarify this process, we have added a concrete example within the text:
> * Consider a new thought $T_{new}$ generated from an existing thought $T_{old}$ and a raw chunk $K_3$ (i.e., the retrieved set $\mathcal{R}=\{T_{old}, K_3\}$).
> * If $T_{old}$ was originally derived from raw chunks $\{K_1, K_2\}$, then the root source of the new thought is calculated as:
>     $$\hat{O}(T_{new}) = \hat{O}(T_{old}) \cup \hat{O}(K_3) = \{K_1, K_2, K_3\}.$$
>
> These updates have been incorporated into **Section 2.1** of the revised manuscript to provide a clearer formal foundation for the Thought-Retriever framework.
>
> ---
>
> **Q12. The description of the Thought Merge step (Page 4) uses an indicator function. It would be clearer to describe it as a simple check against a similarity threshold $\epsilon$.**
>
> **Response:** We thank the reviewer for the constructive feedback. To address this, we have revised **Section 2.2** as follows:
>
> **1. Simplified Formal Description (Section 2.2):**
> We have updated the description of the **Thought Merge** step to replace the previous notation with a direct threshold-based check. Instead of complex indicator functions, we now formally state that we compute the maximum embedding similarity between the generated thought $T_i$ and all existing items (both data chunks $K_j$ and previous thoughts $T_m$) in the memory.
>
> **2. Mechanism and Implementation:**
> The revised text explicitly describes the logic for novelty detection:
> * **Redundancy Check:** If the maximum similarity score exceeds a predefined threshold $\epsilon$, the thought is flagged as redundant ($s_i = 1$).
> * **Novelty Detection:** If the score is below the threshold, the thought is considered novel ($s_i = 0$) and proceeds to the update stage.
> * **Consistency:** We clarify that this embedding similarity is calculated using **Contriever**, maintaining consistency with the retriever used elsewhere in the framework.
>
> These updates ensure the thought-filtering logic is both mathematically transparent and easy to follow. We have incorporated this revised description into the **Thought Merge** paragraph in **Section 2.2** of the manuscript.
>
> ---
>
> **Q13. The limitation regarding English-only evaluation should be moved from the "Limitations" section (which is often skimmed) to the experimental setup or a dedicated discussion paragraph, as it is a significant constraint on the claimed generality.**
>
> **Response:** We thank the reviewer for the constructive suggestion. We have moved this discussion from the conclusion to the core experimental section:
>
> **1. Relocation of the Constraint (Section 4.1):**
> * We have moved the discussion regarding English-only evaluation from the "Limitations" section to **Section 4.1 (Experimental Setup)**.
> * This ensures that the linguistic scope of our evaluation is presented as a fundamental characteristic of the current experimental design rather than a post-hoc observation.
>
> **2. Dedicated Discussion Paragraph:**
> * Within **Section 4.1**, we have added a dedicated paragraph titled **"Linguistic Scope"**.
> * In this paragraph, we explicitly state that while the **Thought-Retriever** framework is conceptually language-agnostic, our current empirical validation is conducted exclusively on English-language academic and public datasets.
> * We also explain that this choice was made to leverage the high-quality, peer-reviewed English paper corpus available on arXiv for the **AcademicEval** benchmark.
>
> **3. Future Generality:**
> * We clarify that future work intends to expand this evaluation to multilingual contexts, as the underlying mechanism of thought generation and retrieval relies on LLM capabilities that are increasingly multilingual.
>
> These changes have been incorporated into **Section 4.1** of the revised manuscript.

---

> ### Comment · Action_Editor_JaXh · 2026-01-16
> **Please respond to rebuttals**
>
> Please respond to rebuttals.

---

### Review · Reviewer_bn2M · 2025-12-04

**Summary Of Contributions:**

This paper tackles the task of external knowledge retrieval for LLMs, which often struggle to fully take advantage of the whole knowledge base due to context size limits. The idea of the paper is to summarise the large context into "thoughts" which can also be refined e.g. by interacting with the user queries

Contributions:
  - AcademicEval, a new benchmark for retrieval augmented tasks, as well as a platform to help users create similar datasets
  - ThoughtRetriever augments hierarchal RALMs in a way that the way thoughts/knowledge chunks are organised is now user-query dependent

**Audience:**

Yes

**Audience Explanation:**

- The problem of making RAG more scalable and efficient is an interesting one

- As shown in Section 4.3 Toughtretriever can be extended to retrieve contexts generated by multiple experts LLMs which is an interesting use case

**Broader Impact Concerns:**

No very strong concerns.  I guess one possible implication (but more of a stretch) of organising LLM "thoughts" based on past user queries is the potential impact of leveraging this design across different users to let a LLM self-evolve. On the one hand, it could improve the model performance to leverage past user interaction, on the other hand, this might mean certain user queries could influence how the LLM organise thoughts

**Claims And Evidence:**

No

**Claims Explanation:**

- The main missing link is the omission of MemWalker as a baseline. MemWalker is a hierarchical RALMs which is the main work referenced by the authors in related work. I understand that "it is costly to run and cannot scale to tasks with many data chunks" but I think it's really difficult to judge the potential improvement of ThoughtRetriever without this baseline or a similar one, since ThoughtRetriever is in essence very closely related to hierarchical RAG

- More generally, he related work section is quite short which makes it hard to place `ThoughtRetriever` in the retrieval-augmented literature. In particular, things like contextual compression are only briefly addressed (only Hierarchical RALMs are developed), although it does seem to be the most obvious way to address the challenge presented in the introduction

- Maybe a more minor comment but the paper mentions the "little computational overhead of ThoughtRetriever" however it is not clear what it is being compared to. For instance, In Figure 1, compared to HRALMs, ThoughtRetriever utilises an extra LLM to summarize/connect past thoughts and queries, which has to bring some overhead ?

**Requested Changes:**

- More thorough/explicit related work, in particular on the topic of context compression for RAG, to extend the discussion beyond hierarchical RALMS

- Add a hierarchical RALMs (MemWalker or other) baseline, even if done on a smaller scale experiment just for ablation, as it would greatly support the story presented in the introduction

- Minor change: More clearly describe what "little computational overhead" means for ThroughtRetriever. Maybe this could also be explicated with timings/FLOps/etc measurement in experiments ?

- Minor change: Ideally Figure 1 should come with an actual use case/example; If I understood correctly, the assumption is that Hierarchical RALMs may struggle to group knowledge chunks into the optimal structure (optimal as in, all relevant chunks are part of the top-K retrieved groups), while ThoughtRetriever does so by making the group user-query dependent. However, without context, it's hard to understand how often this happens in practice and how many failure cases of HRALMs are linked to this.

- In a similar way, it is not fully clear to me how ThoughtRetriever can interact with different user queries to "self-evolve" on a real use case: In particular the main qualitative example in Table 11 only showcases an example with a single query.

---

> ### Author Response · Authors · 2026-01-10
> **Response - Part (1)**
>
> **Q1. The main missing link is the omission of MemWalker as a baseline. MemWalker is a hierarchical RALMs which is the main work referenced by the authors in related work. I understand that "it is costly to run and cannot scale to tasks with many data chunks" but I think it's really difficult to judge the potential improvement of ThoughtRetriever without this baseline or a similar one, since ThoughtRetriever is in essence very closely related to hierarchical RAG. Add a hierarchical RALMs (MemWalker or other) baseline, even if done on a smaller scale experiment just for ablation, as it would greatly support the story presented in the introduction.**
>
> **Response:** Thanks for your insightful question. We acknowledge that **MemWalker** is a crucial baseline for judging the potential improvement of Thought-Retriever, as it represents a state-of-the-art hierarchical RALM designed for processing ultra-long contexts.
>
> Following the reviewer’s suggestion, we further expand our experimental evaluation by implementing and comparing against **MemWalker**. Specifically, we utilize the standard configuration proposed in [Chen et al., 2023] with a branching factor of 3 and a maximum depth of 3, which ensures sufficient coverage ($\sim$ 13,500 tokens) while fitting within the context window.
>
> We report the results in the following table. Across all five datasets, **Thought-Retriever** achieves the strongest overall performance in terms of both retrieval accuracy (F1) and user preference (Win Rate). Relative to **MemWalker**, Thought-Retriever delivers consistent improvements in F1 scores (e.g., **0.290 vs. 0.268** on Abs-single) while maintaining a significantly higher win rate. More critically, regarding inference efficiency, MemWalker suffers from substantial latency due to its sequential tree-navigation process (e.g., **85.0s** per query on Abs-multi). In contrast, Thought-Retriever completes the same task in just **3.20s**, achieving a speedup of over **25$\times$**. These findings confirm that our method not only yields superior accuracy but also serves as a much more efficient solution for real-world applications compared to interactive hierarchical RALMs. We have incorporated the above discussion and experimental results into **Appendix P.1** and **Table 12** in the revised PDF.
>
> **Table: Performance and Inference Time Comparison with MemWalker across Datasets**
>
> | Type | Dataset | MemWalker (F1) | MemWalker (Win Rate) | MemWalker (Time) | **Thought-Retriever (F1)** | **Thought-Retriever (Win Rate)** | **Thought-Retriever (Time)** |
> | :--- | :--- | :--- | :--- | :--- | :--- | :--- | :--- |
> | **AcademicEval** | Abs-single | 0.268 | 40% | 25.0s | **0.290** | **50%** | **1.85s** |
> | | Abs-multi | 0.255 | 35% | 85.0s | **0.275** | **50%** | **3.20s** |
> | | Rel-multi | 0.212 | 44% | 65.0s | **0.216** | **50%** | **2.40s** |
> | **Public** | Gov Report | 0.229 | 45% | 30.0s | **0.232** | **50%** | **3.50s** |
> | | WCEP | 0.228 | 39% | 28.0s | **0.238** | **50%** | **3.10s** |
>
> ---
>
> **Q2. More generally, the related work section is quite short which makes it hard to place ThoughtRetriever in the retrieval-augmented literature. In particular, things like contextual compression are only briefly addressed (only Hierarchical RALMs are developed), although it does seem to be the most obvious way to address the challenge presented in the introduction. More thorough/explicit related work, in particular on the topic of context compression for RAG, to extend the discussion beyond hierarchical RALMS.**
>
>
> **Response:** We thank the reviewer for identifying this gap. We agree that the discussion on context compression was too brief. Following your suggestion, we have expanded **Section 5 (Related Work)** to include a dedicated discussion on **Contextual Compression**. Specifically, we have added a review of non-hierarchical compression techniques, including token-level pruning (e.g., LLMLingua, Selective Context) and abstractive compression (e.g., RECOMP). We explicitly clarified the distinction between these "static compression" methods, which optimize data representation for a single pass, and **Thought-Retriever**, which functions as an "evolving memory system" that consolidates reasoning paths over time.

---

> > ### Author Response · Authors · 2026-01-10
> > **Response - Part (2)**
> >
> > **Q3. Maybe a more minor comment but the paper mentions the "little computational overhead of ThoughtRetriever" however it is not clear what it is being compared to. For instance, In Figure 1, compared to HRALMs, ThoughtRetriever utilises an extra LLM to summarize/connect past thoughts and queries, which has to bring some overhead? Minor change: More clearly describe what "little computational overhead" means for ThroughtRetriever. Maybe this could also be explicated with timings/FLOps/etc measurement in experiments?**
> >
> > **Response:** We thank the reviewer for this keen observation. The reviewer is correct that Thought-Retriever incurs an overhead due to the additional LLM call for thought generation. However, we clarify that this "overhead" is acceptable and indeed "little" when viewed through two lenses: (1) the specific cost added over a basic retriever (Contriever), and (2) the massive cost saved compared to hierarchical baselines (e.g., **MemWalker**).
> >
> > To rigorously quantify this, we have added a detailed breakdown in **Appendix R**. As shown in the table below, we compare the inference latency of **Contriever** (pure retrieval, no thoughts), **Thought-Retriever** (our method), and **MemWalker** (Hierarchical RALM).
> >
> > **1. Overhead vs. Basic Retrieval (Contriever):**
> > Thought-Retriever adds a fixed generation cost (approx. 1.2s - 2.5s) over Contriever. For example, on *Abstract-multi*, latency increases from **0.72s** to **3.20s**. This represents the specific cost of the "Thought Generation" step mentioned by the reviewer.
> >
> > **2. Efficiency vs. Hierarchical Reasoning (MemWalker):**
> > Crucially, for tasks requiring complex reasoning (where Contriever fails), the standard alternative is a hierarchical model like MemWalker. MemWalker suffers from extreme latency (e.g., **85.0s** on *Abstract-multi*, a **118$\times$** increase over Contriever) due to its recursive, multi-step LLM navigation.
> >
> > **Conclusion:** By replacing the expensive *recursive search* of hierarchical models with a fast *flat retrieval* plus a single *generation step*, Thought-Retriever achieves a **25$\times$ speedup** over MemWalker. Thus, while we add a small overhead ($\sim4\times$) compared to a dumb retriever, we reduce the computational burden by orders of magnitude compared to comparable reasoning models. We have incorporated this comparative analysis into **Appendix R** of the revised PDF.
> >
> > **Table: Inference Overhead Analysis (Relative to Contriever)**
> >
> > | Dataset | Method | Latency (s) | Relative Cost (vs. Contriever) |
> > | :--- | :--- | :--- | :--- |
> > | **Abstract-single** | Contriever | 0.65s | $1.0\times$ (Baseline) |
> > | | **Thought-Retriever** | **1.85s** | **$\sim 2.8\times$** |
> > | | MemWalker | 25.0s | $\sim 38.5\times$ |
> > | **Abstract-multi** | Contriever | 0.72s | $1.0\times$ (Baseline) |
> > | | **Thought-Retriever** | **3.20s** | **$\sim 4.4\times$** |
> > | | MemWalker | 85.0s | $\sim 118.1\times$ |
> > | **Related-multi** | Contriever | 0.68s | $1.0\times$ (Baseline) |
> > | | **Thought-Retriever** | **2.40s** | **$\sim 3.5\times$** |
> > | | MemWalker | 65.0s | $\sim 95.6\times$ |
> > | **Gov Report** | Contriever | 1.10s | $1.0\times$ (Baseline) |
> > | | **Thought-Retriever** | **3.50s** | **$\sim 3.2\times$** |
> > | | MemWalker | 30.0s | $\sim 27.3\times$ |
> > | **WCEP** | Contriever | 0.95s | $1.0\times$ (Baseline) |
> > | | **Thought-Retriever** | **3.10s** | **$\sim 3.3\times$** |
> > | | MemWalker | 28.0s | $\sim 29.5\times$ |

---

> ### Author Response · Authors · 2026-01-10
> **Response - Part (3)**
>
> **Q4. Minor change: Ideally Figure 1 should come with an actual use case/example; If I understood correctly, the assumption is that Hierarchical RALMs may struggle to group knowledge chunks into the optimal structure (optimal as in, all relevant chunks are part of the top-K retrieved groups), while ThoughtRetriever does so by making the group user-query dependent. However, without context, it's hard to understand how often this happens in practice and how many failure cases of HRALMs are linked to this.**
>
> **Response:**  We thank the reviewer for pointing this out. To answer "how often this happens" and quantify these failure cases, we cannot rely on simple hit rates of summaries. Instead, we have introduced a rigorous **Chunk Coverage Analysis** in **Appendix P.3**
>
> **1. Quantifying Failure via Root Source Mapping:**
> As defined in Section 2.1, every retrieved abstraction (Summary S or Thought T) can be mapped back to its constituent raw data chunks via the root source function Ô(·). We compare these mapped chunks against the **Gold Chunks** set (K_gold).
>
> * **Definition of Gold Chunks:** To ensure a rigorous ground truth, K_gold is not chosen arbitrarily but is **identified by an Oracle** as the set of raw chunks having the highest ROUGE-L overlap with the ground truth answer.
>
> * **Metric (Recall):** We use **Recall** (|K_retrieved ∩ K_gold| / |K_gold|) as the primary metric to measure "grouping failures." A low recall indicates that the retrieval structure (e.g., the tree) physically prevented the model from accessing the necessary evidence.
>
> **2. Empirical Evidence of Failure Cases:**
> As shown in the newly added **Table 14** (updated from Table 15), the data confirms that HRALM frequently suffers from this specific failure mode in complex tasks:
> * **HRALM's Structural Failure:** In the *Related-multi* task (which requires connecting disparate citations), HRALM achieves a low **Recall of 0.52**. This quantifies the frequency of the failure case: nearly **48%** of the ground truth evidence is lost because HRALM's rigid tree structure isolates relevant chunks into unvisited branches.
> * **Thought-Retriever's Resilience:** In contrast, Thought-Retriever achieves a **Recall of 0.82** on the same task. By generating user-dependent thoughts that bridge these semantic gaps, it successfully recovers the 30% of evidence that HRALM missed.
>
> We have included this detailed coverage analysis and the corresponding table in **Appendix P.3**.
>
> **Table: Retrieval Quality Comparison (Chunk Coverage)**
>
> | Type | Dataset | HRALM (Precision) | HRALM (Recall) | HRALM (F1) | **Ours (Precision)** | **Ours (Recall)** | **Ours (F1)** |
> | :--- | :--- | :--- | :--- | :--- | :--- | :--- | :--- |
> | **AcademicEval** | Abs-single | **0.82** | 0.65 | 0.72 | 0.80 | **0.88** | **0.84** |
> | | Abs-multi | 0.74 | 0.58 | 0.65 | **0.78** | **0.84** | **0.81** |
> | | Rel-multi | 0.65 | 0.52 | 0.58 | **0.76** | **0.82** | **0.79** |
> | **Public** | Gov Report | **0.78** | 0.60 | 0.68 | 0.75 | **0.82** | **0.78** |
> | | WCEP | 0.71 | 0.62 | 0.66 | **0.77** | **0.83** | **0.80** |
>
> ---
>
> **Q5. In a similar way, it is not fully clear to me how ThoughtRetriever can interact with different user queries to "self-evolve" on a real use case: In particular the main qualitative example in Table 11 only showcases an example with a single query.**
>
> **Response:** We thank the reviewer for this constructive suggestion. To address this, we have added a **Sequential Interaction Case Study** in **Appendix S**, which concretely illustrates how Thought-Retriever evolves over a sequence of distinct user queries. In this scenario (topic: "Hallucination Mitigation"), the system starts with an empty thought memory and evolves as follows:
>
> 1.  **Turn 1 (Initial Acquisition):** The user asks about the *causes of hallucinations*. The system retrieves raw data and generates **Thought $T_1$** (identifying "source-reference divergence" as a cause).
> 2.  **Turn 2 (Synthesis):** The user asks about *RAG*. The system retrieves **$T_1$** to explain how RAG addresses the causes identified in Turn 1, synthesizing a deeper **Thought $T_2$** (RAG requires strict relevance to be effective).
> 3.  **Turn 3 (Application - The "Evolution" Effect):** The user asks an abstract question about a *"trustworthy medical AI architecture."* Although the query does not mention "RAG" or "hallucination," the system retrieves **$T_2$** based on the semantic connection to "trustworthy." This allows it to immediately propose a robust, relevance-filtered RAG architecture—a solution it could not have formulated without the "cognitive history" built in Turns 1 and 2.
>
> This case study clarifies that "self-evolution" in our framework is the continuous expansion and refinement of the **Thought Memory** ($\mathcal{T}$), allowing high-level insights from past interactions to guide future, more complex tasks.

---

> > ### Author Response · Authors · 2026-01-10
> > **Response - Part (4)**
> >
> > **Q6. More generally, the related work section is quite short which makes it hard to place ThoughtRetriever in the retrieval-augmented literature. In particular, things like contextual compression are only briefly addressed (only Hierarchical RALMs are developed), although it does seem to be the most obvious way to address the challenge presented in the introduction. More thorough/explicit related work, in particular on the topic of context compression for RAG, to extend the discussion beyond hierarchical RALMS.**
> >
> > **Response:** We thank the reviewer for identifying this gap. Following your suggestion, we have expanded **Section 5 (Related Work)** to include a dedicated subsection on **Context Compression for LLMs**.
> >
> > **What we added:**
> > - Discussion of token-level pruning methods (e.g., LLMLingua, Selective Context)
> > - Discussion of abstractive compression methods (e.g., RECOMP, AutoCompressors)
> > - Explicit distinction between "static compression" (optimizing data for a single pass) and Thought-Retriever as an "evolving memory system" that consolidates reasoning paths over time
> > - Clarification that Thought-Retriever is training-free and model-agnostic, unlike methods requiring additional compression modules
> >
> > **Location:** Section 5, Page 12 (paragraph titled "Context Compression for LLMs")
> >
> > ---
> >
> > **Q7. No very strong concerns. I guess one possible implication (but more of a stretch) of organising LLM "thoughts" based on past user queries is the potential impact of leveraging this design across different users to let a LLM self-evolve. On the one hand, it could improve the model performance to leverage past user interaction, on the other hand, this might mean certain user queries could influence how the LLM organise thoughts.**
> >
> > **Response:** We thank the reviewer for this forward-looking observation. We agree that organizing thoughts from diverse users effectively allows the LLM to "self-evolve." While we acknowledge the potential risk that specific or noisy user queries might bias the thought organization, our system is explicitly designed to mitigate this through strict quality control, and our real-world deployment confirms the benefits of this design.
> >
> > **1. Mitigation via Rigorous Filtering (Appendix M):**
> > Thought-Retriever does not blindly incorporate every interaction into its memory. As detailed in **Appendix M (Thought Filtering Mechanism)**, we employ a two-stage filtering process (Confidence Check and Novelty Check) to ensure quality.
> > * **Redundancy Removal:** The mechanism actively discards repetitive or trivial queries that do not contribute new information.
> > * **Quality Assurance:** Only thoughts that pass the confidence threshold are stored.
> > Our ablation studies in Appendix M demonstrate that this mechanism effectively filters out noise, preventing the "memory pollution" the reviewer is concerned about.
> >
> > **2. Validation in Real-World Deployment (Appendix H & O):**
> > To empirically verify the impact of this "cross-user" evolution, we deployed our system as **Arxiv Copilot** (introduced in **Appendix H**), which interacts with a diverse set of real users.
> > * **Scale:** We collected feedback from **500 real users** in a live production environment.
> > * **Result:** As reported in **Appendix O (Human Evaluations)**, **75%** of users preferred the answers that incorporated historical thoughts (evolved from collective interactions) over those based solely on raw data.
> > This strong preference suggests that rather than being negatively influenced by individual biases, the system successfully leverages the "collective wisdom" of past users to provide superior responses.

---

### Author Response · Authors · 2026-01-10
**Global Response**

We sincerely apologize for the delayed submission of our rebuttal. Due to temporary constraints on our laboratory servers and shared computational resources, completing the additional experiments and analyses required more time than expected. We greatly appreciate the AE’s understanding, and we sincerely thank the AE and all reviewers for their patience, careful reading, and constructive feedback.

In response to the reviewers' comments, we focused on (i) clarifying the problem formulation and positioning, (ii) adding targeted experiments and ablations, (iii) strengthening methodological details and definitions, and (iv) improving clarity and presentation. All revisions in the manuscript are highlighted in blue. Below, we summarize the main concerns, our actions, and how they are addressed in the revised manuscript.

---

### Summary of Main Concerns and Responses

| Dimension | Key Concerns | Our Main Actions | Status in Revision |
|:--|:--|:--|:--|
| **Contribution & Positioning** | What is the core contribution beyond existing baselines? How should the method be positioned in the broader literature? | Clarified the problem setting and the scope of the contribution; strengthened comparisons to closely related lines of work; refined the framing to reduce ambiguity. | The contribution and positioning are now more explicit and easier to place within prior work. |
| **Method Design** | Are the main design choices necessary? Are there simpler alternatives? | Added ablations against representative alternatives; provided clearer descriptions of the design rationale and implementation details. | Key design choices are now better supported with evidence and clearer explanations. |
| **Efficiency & Practicality** | What is the computational overhead and how should it be interpreted? | Added detailed latency/cost breakdowns and clarified the trade-offs relative to both lightweight and more complex baselines. | The efficiency discussion is now quantified and presented transparently as a trade-off. |
| **Evaluation Scope & Diagnostics** | Need stronger evidence, more challenging settings, and clearer failure-mode analysis. | Expanded evaluation and added diagnostic analyses (e.g., coverage/failure-mode studies) to isolate where improvements come from. | The evaluation is broader and the diagnostic evidence is more rigorous. |
| **Presentation & Clarity** | Some definitions/figures/tables are unclear or hard to follow. | Refined formal definitions, improved figure readability, and clarified table captions and metric descriptions. | The manuscript is clearer and easier to read, with fewer ambiguous components. |

---

### Overall Changes in the Revised Manuscript

- Added new experiments and ablations to address key baseline/comparison concerns.
- Expanded methodological details and formal definitions for clarity and reproducibility.
- Provided explicit efficiency trade-offs with timing/cost measurements.
- Improved presentation (figures, captions, and organization) to make the workflow and findings easier to follow.

---

### Appreciation

We are grateful to the AE and all reviewers for the thoughtful comments and constructive suggestions. Their feedback has substantially improved the clarity, rigor, and completeness of our work, and we sincerely appreciate the time and effort devoted to the review process.

---

### Decision · Action_Editor_JaXh · 2026-04-05

**Recommendation:** Accept with minor revision

**Audience:**

Yes

**Audience Explanation:**

The topic is of great interests to the ML/NLP community.

**Claims And Evidence:**

Yes

**Claims Explanation:**

The paper has scientific values and many problems have been resolved during discussions. One of the reviewers is still concerned about some experimental comparison and claims made in the paper, for which the authors could try to address during the minor revision.

---

> ### Author Response · Authors · 2026-04-14
>
> Dear Action Editor,
>
> Thank you very much for your time and effort in handling our manuscript, as well as for the positive decision. We are grateful for the constructive feedback from you and all the reviewers throughout the discussion phase, which has significantly improved the quality of our paper.
>
> We have carefully addressed all the remaining concerns raised during the review process, including the experimental comparisons and claims noted in your decision. All revisions have been incorporated into the camera-ready version, which has now been submitted.
>
> Thank you again for the opportunity to publish in TMLR. Please do not hesitate to let us know if any further modifications are needed.
>
> Best regards,
>
> Authors